# CROSS-MODAL ALIGNMENT VIA VARIATIONAL COPULA MODELLING

## ABSTRACT

Various data modalities are common in real-world applications. In healthcare, for example, electronic health records, medical images, and clinical notes provide comprehensive information for diagnosis and treatment. Thus, it is essential to develop multimodal learning methods that aggregate information from multiple modalities to generate meaningful representations for downstream tasks. The key challenge here is how to appropriately align the representations of the respective modalities and fuse them into a joint distribution. Existing methods mainly focus on fusing the representations via concatenation or the Kronecker product, which oversimplifies the interaction structure between modalities, prompting the need to model more complex interactions. Moreover, the notion of joint distribution of the latent representation that incorporates higher-order interactions between modalities is also underexplored. Copula is a powerful statistical structure in modelling the interactions between variables, as it bridges the joint distribution and marginal distributions of multiple variables. In this paper, we propose a novel copula modelling-driven multimodal learning framework, which focuses on learning the joint distribution of various modalities to capture the complex interaction among them. The key idea is interpreting the copula model as a tool to align the marginal distributions of the modalities efficiently. By assuming a Gaussian mixture distribution for each modality and a copula model on the joint distribution, our model can also generate accurate representations for missing modalities. Extensive experiments on public MIMIC datasets demonstrate the superior performance of our model over other competitors. Ablation studies also validate the effectiveness of the copula alignment strategy and the robustness of our model over different choices of the copula family. Code is anonymously available at https://anonymous.4open.science/r/CM2-C1FD/README.md.

## 1 INTRODUCTION

Multimodal learning aims to aggregate information from multiple modalities to generate meaningful representations for downstream tasks. It has been widely explored in the context of vision-language models (Fu et al., 2023; El Banani et al., 2023), audio-visual applications (Chen et al., 2023; Mo & Tian, 2023; Huang et al., 2023), image-video models (Girdhar et al., 2023; Gan et al., 2023) and healthcare applications (Wu et al., 2024a; Hayat et al., 2022). For example, multimodal learning has been applied to various healthcare tasks such as clinical prediction tasks (Zhang et al., 2023; Wu et al., 2024a), report generation (Song et al., 2022; Cao et al., 2023), and clinical trial site selection (Theodorou et al., 2024). The existing fusion strategies can be divided into early, joint, or late fusion (Huang et al., 2020), where the joint fusion paradigm is the most popular strategy and its core idea is to model the interactions between the representations of the input modalities (Hayat et al., 2022). The resulting fused embedding encodes the structural interaction between the modalities, enabling accurate prediction for each modality.

However, due to the heterogeneity of different modalities (e.g., medical images, medical reports, EHRs), properly aligning the distributions of the various modalities remains a challenging problem. The existing alignment strategies mainly rely on concatenation or Kronecker products which oversimplify the interaction between different modalities. A recent work Salzmann et al. (2022) emphasizes simple probabilistic assumptions on the marginals and neglects to explore statistical assumptions about the joint distributions of the modalities. This approach may result in biased fused representa-

tions, limiting the performance of downstream tasks and the generalizability and robustness of the resulting multimodal models. Therefore, there is still a need for an approach that more appropriately aligns the distributions of modalities and models the potentially complex interactions between them.

Copula models have shown great success in modelling the interactions of variables as they construct a bridge between the joint distribution and their marginals (Cherubini, 2004). However, copula models are less explored in the deep learning field as most of the existing approaches heavily rely on sampling-based methods (e.g., MCMC (Silva & Gramacy, 2009)), which are relatively slow and difficult to scale to modern deep learning settings (Smith & Loaiza-Maya, 2023). Although some recent works are attempting to introduce copula to deep learning models through stochastic variational inference (Smith & Loaiza-Maya, 2023), the potential of copula in multimodal learning is still underexplored.

Moreover, existing multimodal learning methods mostly assume the existence of all modalities. In reality, some modalities may be missing for some observations due to various reasons, e.g., missing medical images or reports for some patients due to clinical and administrative factors in healthcare, which pose a major challenge in multimodal learning. The existing solutions either discard these observations or impute simple values (e.g., zeros or means from other observations) to address the missing modality problem. However, these approaches ignore the marginal distributions of the modality and often mislead the learning of the joint distribution. Therefore, properly learning the marginal distribution is also necessary to generate unbiased representations for the observations with missing modalities.

In light of the above challenges, we propose a novel copula modelling-driven multimodal learning framework, namely $\text{CM}^2$ (**C**ross-**M**odal alignment via variational **C**opula **M**odelling), to tackle the joint fusion paradigm from a probabilistic perspective. Our contributions can be summarized as: (1) We for the first time introduce copula modelling into multimodal learning, where we interpret copula as an effective tool of distribution alignment, guaranteed by Sklar's theorem. (2) We employ a Gaussian mixture model on the marginal distribution of each modality to enable more flexible modelling of the high-dimensional feature distribution of different modalities. (3) We adopt stochastic variational inference to optimize the copula model, which enables the scalability of our model to large-scale datasets. (4) We adopt the learned marginal distribution as the data generator to accurately impute the missing observations. (5) Empirical results on real multimodal MIMIC datasets demonstrate the good performance of our method and ablation analysis corroborates the effectiveness of copula in modality alignments and robustness to potential variations.

## 2 RELATED WORKS

**Multimodal Representation Learning.** Multimodal representation learning aims to effectively integrate information from different modalities for accurate predictions on the downstream tasks. Early works (Hayat et al., 2022; Ding et al., 2022; Trong et al., 2020) focus on late fusion that merges unimodal representations via, for instance, concatenation or Kronecker product. However, such approaches oversimplify the interactions of the modalities and mostly lead to biased fused representations. Therefore, the structural interactions of the modalities need to be encoded in the fused representation for more effective multimodal learning. Recently, modelling the interaction between modalities has received increasing attention. Liang et al. (2024) proposed an information decomposition framework to define and quantify different types of interactions between modalities. Transformer-based methods have greatly facilitated the progress by modelling the cross-model tokens (Zhang et al., 2023; Theodorou et al., 2024). However, matching the correspondence with transformers introduces high computational complexity, which prompts a more efficient approach for representation alignment.

**Copula Deep Learning.** Copula is a promising tool in modelling the interactions or correlations between variables and it constructs a bridge between the joint distribution and marginal distributions. Copula has been widely applied in financial risk management (Hofert, 2021; Rodriguez, 2007), signal processing, and healthcare (Zeng & Wang, 2022) due to its capability in modelling complex interactions. Traditional copula models rely on closed-form solutions of the likelihood and estimate the copula parameter with sampling-based approaches (e.g., MCMC (Silva & Gramacy, 2009)). However, these algorithms suffer from high time complexity, making them less applicable to high-

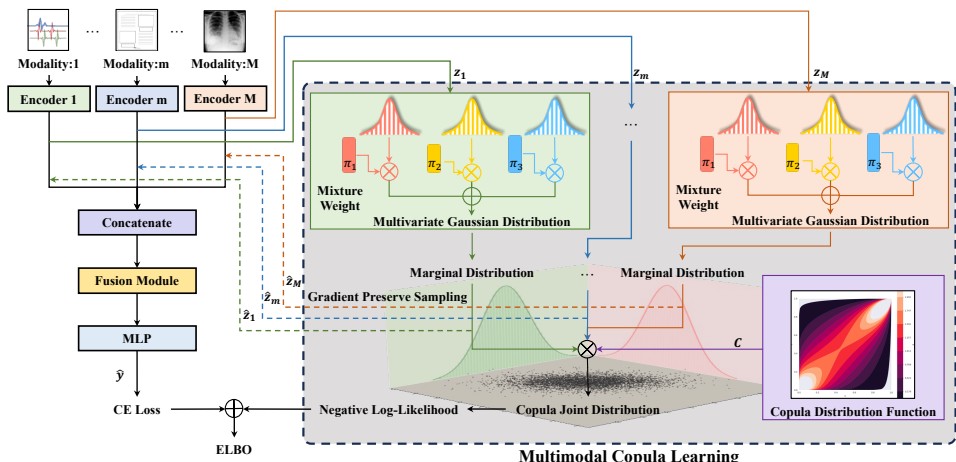

Figure 1: Overview of our proposed $\text{CM}^2$ framework. For a dataset with $M$ modalities, we extract modality-specific embeddings $z_m$ via Encoder$_m$ and compute its GMM. We then model the marginal distribution and estimate the joint distribution using a copula family $C$. If modality $m$ is missing, we sample $\hat{z}_m$ from its GMM. The concatenated embedding $z$ then passes through a 2-layer LSTM fusion module and MLP classifier to predict $\hat{y}$. The ELBO for backpropagation can be obtained by aggregating the task-specific loss (e.g., CE loss) and the negative log-likelihood from the joint distribution.

dimensional data. Recently, with the emergence of deep learning, there have been works integrating copula models into deep learning frameworks (Tagasovska et al., 2019; Smith et al., 2020). To tackle the inherent high dimensionality, variational inference is adopted to solve copula models in high dimensions (Tran et al., 2015; Smith & Loaiza-Maya, 2023). For example, Tagasovska et al. (2019) introduced copula to variational autoencoders to create deep generative models. However, the potential of copula in multimodal learning is still under-explored.

**Learning with Missing Data.** Traditional multimodal learning assumes all modalities are available, but in reality, some observations may be missing, e.g., missing medical images or reports in clinical data. Late fusion is a common strategy to address missing modalities by aggregating predictions (Yoo et al., 2019) or latent space representations (Theodorou et al., 2024) from the available modalities. Although it is effective, it treats each modality independently and lacks interaction between them. Some researches focus on extracting shared information across modalities to perform downstream tasks (Deldari et al., 2023; Yao et al., 2024). However, learning such shared representations can be challenging, particularly when the modalities are highly heterogeneous, as in the case of EHRs and CXRs. Many approaches attempt to preserve model performance via modelling the relationships between them (Zhang et al., 2022; Wu et al., 2024b) or generating a global representation for the missing data (Hayat et al., 2022). Other methods assume that the missing modality follows a certain distribution, imputing the missing values using the mean or mode of that distribution (Ma et al., 2021). Despite their successes, these distributional assumptions may be inaccurate, potentially introducing bias into the model. Therefore a probabilistic assumption is needed to guarantee the unbiasedness of learned marginal distributions.

## 3 METHODOLOGY

### 3.1 PRELIMINARIES

**Copula.** A $M$-variate function $C(u_1, \ldots, c_M)$ where $u_m \in [0, 1]$ for all $i$ is a copula if and only if $C$ defines a valid joint cdf of the random vector $(U_1, \ldots, U_M)$ with each $U_m$ distributed as uniform on the unit interval. Without loss of generality, we select the Gumbel copula for illustration as it is

based on the extreme values of each modality, which can best represent the strongest signals in each modality. Given $u$ and $v$ the c.d.f. values of the first and second modality, respectively, the bivariate form of Gumbel copula is defined by

$$C(u, v; \alpha) = \exp\{-[(-\log u)^\alpha + (-\log v)^\alpha]^{\frac{1}{\alpha}}\},$$

and its copula density is

$$c(u, v; \alpha) = \frac{1}{uv}(-\log v)^{\alpha-1}(-\log u)^{\alpha-1}C(u, v; \alpha)\left[g(u, v; \alpha)\right]^{\frac{2(1-\alpha)}{\alpha}}\left[(\alpha-1)\left[g(u, v; \alpha)\right]^{-\frac{1}{\alpha}} + 1\right],$$

where $g(u, v; \alpha) = (-\log u)^\alpha + (-\log v)^\alpha$. The effects of different Copula families are discussed in the ablation analysis. Details of different copula families and their corresponding distribution and density functions are provided in the Appendix C.

**Multimodal Learning.** Given the multimodal training dataset $\mathcal{D}_{\text{tr}} = \{(\boldsymbol{x}_1^{(i)}, \ldots, \boldsymbol{x}_M^{(i)}, y^{(i)})\}_{i=1}^n$, where $\boldsymbol{x}_m^{(i)}$ is the $i$-th observation of the $m$-th modality and $y^{(i)}$ is the corresponding label, the goal is to train a multimodal model $f_\Theta(\cdot)$ with parameter $\Theta$ such that the model can achieve optimal performance in downstream tasks.

## 3.2 Copula Multimodal Learning

The overview of the proposed copula-driven multimodal learning framework is shown in Figure 1. Given multimodal data, we extract each modality-specific embedding and compute its Gaussian mixture model (GMM). We then model the marginal densities and estimate the joint distribution using a copula family $C$. If modality $m$ is missing, we generate feature embeddings from its GMM. The concatenated embeddings $\boldsymbol{z}$ are passed through a fusion module and an MLP classifier for prediction. The ELBO combines the copula log-likelihood and task-specific loss.

**Gaussian Mixture Assumption.** To generate a more flexible feature distribution, we assume the feature distribution of the $m$-th modality follows a $K$-mixture of multivariate GMM,

$$f_m(\boldsymbol{z}) = \sum_{k=1}^K \pi_{mk}\mathcal{N}(\boldsymbol{\mu}_{mk}, \boldsymbol{\Sigma}_{mk}), \tag{1}$$

where $\pi_{mk}$ is the mixture weight, $\boldsymbol{\mu}_{mk}$ is the mean vector, and $\boldsymbol{\Sigma}_{mk}$ is the covariance matrix of the $k$-th mixture of the $m$-th modality. We let $\boldsymbol{\mu} = \{\boldsymbol{\mu}_{mk} : m \in [M], k \in [K]\}$ and $\boldsymbol{\Sigma} = \{\boldsymbol{\Sigma}_{mk} : m \in [M], k \in [K]\}$. Without loss of generality, we predict $\pi_{mk}$ with an MLP with a softmax output layer and adopt the reparameterization trick (Nalisnick, 2018; Tran et al., 2022) which assumes $\boldsymbol{\Sigma}_{mk}$ is diagonal. We further set $\boldsymbol{\mu}$ and $\boldsymbol{\Sigma}$ to be trainable by gradient backpropagation. We compute the cumulative distribution function of the multivariate Gaussian distributions using the approximation provided in Marmin et al. (2015). By employing a mixture model, we can model a wider range of distributions of each modality and improve the flexibility and robustness.

**Multivariate Copula.** We model the joint distribution of the modalities with multivariate copula. Using the multivariate copula, the joint distribution function of the modality can be written as

$$F_{z_1,\ldots,z_M}(z) = C(F(z_1), \ldots, F(z_M)),$$

where $C(F(z_1), \ldots, F(z_M))$ is the $M$-dimensional copula distribution function, and $F_m(z)$ is the marginal cumulative distribution function of the $m$-th modality which is the c.d.f. of the GMM model defined in Eq. (1).

## 3.3 Stochastic Variational Inference

To tackle the scalability of $\text{CM}^2$ to modern deep learning settings, we adopt the stochastic variational inference to optimize the proposed copula model and treat the copula parameter $\alpha$ as trainable. Algorithm 1 presents the overall workflow of our method.

**Variational Family.** We use a variational posterior $q$ to approximate the true posterior of the joint distribution. The variational family of the copula model that we optimize during training is given by

$$q(z) = \left[\prod_{m=1}^M q_m(z)\right] c(Q_1(z), \ldots, Q_m(z)),$$

---

**Algorithm 1** Sampling algorithm of our proposed framework.

**Input:**
Multimodal model $f_\Theta(\cdot)$ with parameter $\Theta$
The Copula parameter $\alpha$
1: Means and covariances $\{(\boldsymbol{\mu}_{mk}, \boldsymbol{\Sigma}_{mk}) \,|\, \forall m = 1, \ldots, M, k = 1, \ldots, K\}$
2: Multimodal training dataset $\mathcal{D}_{\mathrm{tr}} = \{(\boldsymbol{x}_1^{(i)}, \ldots, \boldsymbol{x}_M^{(i)}, y^{(i)})\}_{i=1}^n$
   **Output:** Trained $f_\Theta$
3: **for** $(\boldsymbol{x}_1^{(i)}, \ldots, \boldsymbol{x}_M^{(i)}, \boldsymbol{y}^{(i)})$ in $\mathcal{D}_{tr}$ **do**
4:     $\hat{y}^{(i)} = f_\Theta(\boldsymbol{x}_1^{(i)}, \ldots, \boldsymbol{x}_M^{(i)})$                                ▷ Forward propagation
5:     Compute task-specific loss $\mathcal{L}_{\mathrm{obj}}$ with $\hat{y}^{(i)}$ and $y^{(i)}$
6:     Compute the $\mathrm{KL}(q\|\pi)$ and hence the ELBO
7:     Backpropagate the ELBO to update $\Theta$
8: **end for**
9: **Return:** Trained $f_\Theta$

---

where $q_m(z)$ is the density of the variational posterior of the GMM of the $m$-th modality, and $Q_m(z)$ is the corresponding c.d.f.

**The Evidence Lower Bound (ELBO).** The joint objective function can be written as the negation of the negative log-likelihood

$$\text{ELBO} = -\lambda_{\mathrm{cop}} \sum_{i=1}^n \left( \log c(Q_1(z_i), \ldots, Q_m(z_i)) - \sum_{m=1}^M \log f_m(z_i) \right) + \mathcal{L}_{\mathrm{obj}},$$

where $c(Q_1(z), \ldots, Q_m(z))$ is the copula density, $\lambda_{\mathrm{cop}}$ is the regularization parameter of the copula assumption, and $\mathcal{L}_{\mathrm{obj}}$ is the task-specific loss (e.g., the cross-entropy loss). We compute the gradient based on the ELBO and backpropagate it to $\boldsymbol{\mu}$ and $\boldsymbol{\Sigma}$ to learn the marginal distributions of each modality, together with the copula parameter $\alpha$ to learn the interactions between these modalities and the multimodal model parameter $\Theta$ to learn the embedding, fusion, and classification layers.

### 3.4 HANDLING MISSING MODALITY

Thanks to the probabilistic design of our method, our framework can also generate pseudo observations for missing modalities. Without loss of generality, we consider missing modalities with complete labels where only the observations are missing. The learned GMM for each modality can be treated as a data generation model, and we can generate feature embeddings through sampling from the GMM of each modality (i.e., $\boldsymbol{z}_m^{(i)} \sim F_m$). Then the generated feature embeddings can be treated as the future input to the classification layer and predictions can be obtained.

By learning the copula parameter $\alpha$, the marginal distribution on each modality contains information from other modalities and information of the interactions. The generated feature representation $\boldsymbol{x}_m^{(i)}$ can therefore better reflect the characteristics of the joint distribution, which would improve the quality of the representation and the downstream task performance as a result.

### 3.5 THEORETICAL GUARANTEE WITH SKLAR'S THEOREM.

We make use of Sklar's theorem to demonstrate the uniqueness of the joint distribution. Sklar's theorem is given as follows.

**Theorem 1.** *(Sklar's theorem) (Sklar, 1959) Let $F(x_1, \ldots, x_M)$ be an $M$-variate c.d.f. for $(X_1, \ldots, X_M)$ with the marginal c.d.f. for the $m$-th variable given by $F_m(x_m), m = 1, \ldots, M$. Then:*

    *1. There exists an $M$-dimensional copula such that*

$$C(F_1(x_1), \ldots, F_M(x_M)) = F(x_1, \ldots, x_M) \tag{2}$$

    *for all $x_m \in \mathbb{R}$.*

2. *Conversely, given any copula $C$ and univariate c.d.f.s $F_1, \ldots, F_M$, $C$ is a valid joint c.d.f. for $(X_1, \ldots, X_M)$. Furthermore, if $F$ is continuous, then $C$ in Eq. (2) is unique.*

The above theorem allows us to construct joint distributions with the same margin but different dependence structures, or conversely by fixing the dependence structure and variating the behaviour in individual modalities (Tagasovska et al., 2019). This allows us to update the marginal distributions and the copula parameter separately. Furthermore, since we assume a GMM for each modality and they are continuous by definition, the uniqueness of the copula $C$ can be guaranteed and the identifiability of the model can be enhanced.

## 4 EXPERIMENTS

### 4.1 DATASETS AND EXPERIMENTAL SETTING

**Datasets.** We evaluate the performance of $\text{CM}^2$ using large-scale, real-world EHR datasets: MIMIC-III (Johnson et al., 2016), MIMIC-IV (Johnson et al., 2023), and MIMIC-CXR (Johnson et al., 2019). MIMIC-III and MIMIC-IV are publicly available datasets containing real-world EHR data from patients admitted to the intensive care units (ICUs) or emergency departments of Beth Israel Deaconess Medical Center (BIDMC), comprising numerical time series and clinical notes. MIMIC-CXR is a public dataset of Chest X-ray(CXR) images along with radiology reports collected from BIDMC, with a subset of patients matched to those in MIMIC-IV.

Following (Hayat et al., 2022), we utilize the MIMIC-IV and MIMIC-CXR datasets for our multi-modal experiments. Additionally, we extend our experiments to the MIMIC-III dataset. As CXR images are not available in MIMIC-III, we replace them with clinical notes serving as the second modality. Table 1 provides an overview of the datasets used in our experiments. We extracted 25,071 ICU stays with EHR records from MIMIC-IV, 5,931 of which are matched to CXR images and reports. Similarly, we extracted 21,139 ICU stays with EHR records from MIMIC-III, with 5,273 stays matched to clinical notes. To evaluate the performance of $\text{CM}^2$ on cross-modal alignment, we conduct experiments on *totally matched* bi-modal and tri-modal settings. We also evaluate the performance on *partially matched* datasets to demonstrate the robustness of $\text{CM}^2$ in the presence of missing modalities. Further details on the datasets can be found in the appendix A.1.

**Task & Evaluation Metrics.** Following the common practice in clinical prediction tasks (Hayat et al., 2022; Zhang et al., 2022; Wu et al., 2024b; Wang et al., 2024), we focus on two common clinical prediction tasks: (1) **In-Hospital Mortality (IHM)** prediction, which predicts whether a patient will pass away during their hospital stay; and (2) **Readmission (READM)** prediction, which

Table 1: Datasets description

| Dataset | No. Train | No. Valid | No. Test | No. Total |
|---|---|---|---|---|
| *Complete Datasets.* | | | | |
| **MIMIC-III** | 14,681 | 3,222 | 3,236 | 21,139 |
| **MIMIC-III NOTE** | 3,652 | 815 | 806 | 5,273 |
| **MIMIC-IV** | 18,064 | 2,035 | 4,972 | 25,071 |
| **MIMIC-CXR** | 344,529 | 9,497 | 23,069 | 377,095 |
| *Matched Datasets.* | | | | |
| **MIMIC-III \| NOTE** | 3,652 | 815 | 806 | 5,273 |
| **MIMIC-IV \| CXR** | 4,287 | 465 | 1,179 | 5,931 |
| **MIMIC-IV \| CXR \| REPORT** | 4,287 | 465 | 1,179 | 5,931 |

aims to predict whether a patient will be readmitted within 30 days after discharge. Both tasks are formulated as binary classification problems. To assess model performance, we compute the area under the precision-recall curve (AUPR) and the area under the receiver operating characteristic Curve (AUROC). Results are reported with the corresponding 95% confidence intervals, obtained through 1,000 bootstrap iterations.

**Backbone Encoders.** Following (Hayat et al., 2022), we utilize ResNet34 (He et al., 2016) as the backbone encoder for CXR image data. For time-series data, including lab values and vital signs, we employ a two-layer stacked LSTM network (Graves & Graves, 2012). For clinical notes and radiology reports, we use the TinyBERT encoder (Jiao et al., 2019). Additionally, a projection layer is applied to map the modality embeddings into the same latent space.

Table 2: Result on MIMIC-III and MIMIC-IV datasets with *totally matched* modalities. All results are reported in AUROC and AUPR with 95% confidence intervals. The best results are highlighted in **bold**. Our proposed method $\text{CM}^2$ outperforms the baselines in all cases.

| Model | IHM | | READM | |
|---|---|---|---|---|
| | AUROC ($\uparrow$) | AUPR ($\uparrow$) | AUROC ($\uparrow$) | AUPR ($\uparrow$) |
| **MIMIC-III** | | | | |
| MMTM (Joze et al., 2020) | $0.776_{(0.728,\ 0.819)}$ | $0.347_{(0.268,\ 0.447)}$ | $0.716_{(0.670,\ 0.762)}$ | $0.341_{(0.277,\ 0.419)}$ |
| DAFT (Pölsterl et al., 2021) | $0.792_{(0.746,\ 0.839)}$ | $0.388_{(0.299,\ 0.484)}$ | $0.701_{(0.653,\ 0.746)}$ | $0.325_{(0.262,\ 0.403)}$ |
| Unified (Hayat et al., 2021) | $0.827_{(0.782,\ 0.868)}$ | $0.466_{(0.371,\ 0.569)}$ | $0.714_{(0.662,\ 0.759)}$ | $0.423_{(0.344,\ 0.504)}$ |
| MedFuse (Hayat et al., 2022) | $0.826_{(0.781,\ 0.866)}$ | $0.430_{(0.340,\ 0.537)}$ | $0.725_{(0.676,\ 0.774)}$ | $0.414_{(0.338,\ 0.502)}$ |
| DrFuse (Yao et al., 2024) | $0.835_{(0.793,\ 0.874)}$ | $0.511_{(0.417,\ 0.607)}$ | $0.749_{(0.699,\ 0.795)}$ | $0.441_{(0.356,\ 0.527)}$ |
| $\text{CM}^2$ | $\mathbf{0.854}_{(0.820,\ 0.861)}$ | $\mathbf{0.513}_{(0.460,\ 0.557)}$ | $\mathbf{0.754}_{(0.731,\ 0.774)}$ | $\mathbf{0.445}_{(0.403,\ 0.487)}$ |
| **MIMIC-IV** | | | | |
| MMTM (Joze et al., 2020) | $0.802_{(0.770,\ 0.835)}$ | $0.429_{(0.362,\ 0.513)}$ | $0.713_{(0.677,\ 0.750)}$ | $0.420_{(0.362,\ 0.489)}$ |
| DAFT (Pölsterl et al., 2021) | $0.815_{(0.782,\ 0.844)}$ | $0.454_{(0.387,\ 0.538)}$ | $0.729_{(0.692,\ 0.766)}$ | $0.433_{(0.378,\ 0.499)}$ |
| Unified (Hayat et al., 2021) | $0.808_{(0.778,\ 0.840)}$ | $0.429_{(0.367,\ 0.512)}$ | $0.719_{(0.680,\ 0.756)}$ | $0.450_{(0.390,\ 0.513)}$ |
| MedFuse (Hayat et al., 2022) | $0.813_{(0.777,\ 0.844)}$ | $0.448_{(0.380,\ 0.528)}$ | $0.725_{(0.690,\ 0.762)}$ | $0.438_{(0.379,\ 0.508)}$ |
| DrFuse (Yao et al., 2024) | $0.818_{(0.784,\ 0.850)}$ | $0.460_{(0.391,\ 0.540)}$ | $0.726_{(0.689,\ 0.760)}$ | $0.430_{(0.370,\ 0.495)}$ |
| $\text{CM}^2$ | $\mathbf{0.827}_{(0.790,0.859)}$ | $\mathbf{0.492}_{(0.423,0.566)}$ | $\mathbf{0.737}_{(0.704,\ 0.773)}$ | $\mathbf{0.466}_{(0.404,\ 0.529)}$ |

## 4.2 COMPARED METHODS

We compare $\text{CM}^2$ against the following baselines: (1) **MMTM** (Joze et al., 2020) is a flexible plugin module that facilitates information exchange between modalities. Since the model assumes full modality availability, we compensate for missing CXR and clinical notes during training and testing by filling in all zeros. (2) **DAFT** (Pölsterl et al., 2021) is a module designed to exchange information between tabular data and image modalities when integrated into CNN models. Similarly, we replace missing CXR and clinical notes with zero matrices during training and testing. (3) **Unified** (Hayat et al., 2021) is a dynamic approach for integrating auxiliary data modalities, learning modality-specific representations, and combining them via a unified classifier. It handles missing data inherently and leverages all available modality-specific information. (4) **MedFUSE** (Hayat et al., 2022) employs LSTM-based fusion to combine features from image or language encoders with EHR encoders. It handles missing modalities by learning a global representation for absent CXR or clinical notes. (5) **DrFuse** (Yao et al., 2024) leverages disentangled representation learning to create a shared representation between the EHR and image modalities, even when one modality is missing.

## 4.3 EXPERIMENTAL RESULTS

**Quantitative Results.** Table 2 presents results on the MIMIC-III and MIMIC-IV datasets with *totally matched* modalities. $\text{CM}^2$ outperforms all baselines in all cases. Notably, for the IHM task, $\text{CM}^2$ exceeds the best baseline by 1.9% in AUROC on MIMIC-III and 3.2% in AUPR on MIMIC-IV. These results demonstrate the effectiveness of $\text{CM}^2$ in capturing the interactions between modalities and enhancing the performance of multimodal learning tasks in clinical prediction.

Table 3 reports results on the MIMIC-III and MIMIC-IV datasets with *partially matched* modalities(e.g. missing modality). $\text{CM}^2$ outperforms the baselines in all cases, with the best performance on the MIMIC-III dataset, where it outperforms the best baseline by 1.5% in AUPR for the IHM task and 0.8% in AUPR for the READM task. This indicates that $\text{CM}^2$ effectively learns the joint distribution of the modalities, generating robust and unbiased representations in the presence of missing modalities.

Moreover, our results reveal that the performance on the partially matched datasets is superior to that on the matched datasets. This can be attributed to the larger number of observations in the partially matched datasets, underscoring the importance of multimodal learning in the presence of missing modalities. Lastly, we observe that the performance on MIMIC-IV is better than that on MIMIC-III under partially matched setting, likely due to the larger number of observations in MIMIC-IV. Additionally, the heterogeneity between modalities in MIMIC-IV may be greater than in

Table 3: Result on MIMIC-III and MIMIC-IV datasets with *partially matched* modalities (i.e., missing modalities). All results are reported in AUROC and AUPR with 95% confidence intervals. The best results are highlighted in **bold**. Our proposed method CM$^2$ outperforms the baselines in all cases.

| Model | IHM | | READM | |
|---|---|---|---|---|
| | AUROC (↑) | AUPR (↑) | AUROC (↑) | AUPR (↑) |
| **MIMIC-III** | | | | |
| MMTM (Joze et al., 2020) | $0.846_{(0.825, 0.865)}$ | $0.450_{(0.399, 0.509)}$ | $0.742_{(0.716, 0.766)}$ | $0.413_{(0.371, 0.455)}$ |
| DAFT (Pölsterl et al., 2021) | $0.854_{(0.836, 0.873)}$ | $0.495_{(0.440, 0.552)}$ | $0.748_{(0.724, 0.772)}$ | $0.429_{(0.386, 0.473)}$ |
| Unified (Hayat et al., 2021) | $0.849_{(0.829, 0.868)}$ | $0.491_{(0.436, 0.542)}$ | $0.751_{(0.728, 0.772)}$ | $0.427_{(0.383, 0.467)}$ |
| MedFuse (Hayat et al., 2022) | $0.850_{(0.830, 0.868)}$ | $0.480_{(0.426, 0.533)}$ | $0.753_{(0.730, 0.775)}$ | $0.437_{(0.396, 0.480)}$ |
| DrFuse (Yao et al., 2024) | $0.839_{(0.817, 0.861)}$ | $0.474_{(0.422, 0.531)}$ | $0.749_{(0.727, 0.770)}$ | $0.411_{(0.371, 0.455)}$ |
| CM$^2$ | $\mathbf{0.856}_{(0.833, 0.877)}$ | $\mathbf{0.510}_{(0.463, 0.566)}$ | $\mathbf{0.754}_{(0.708, 0.795)}$ | $\mathbf{0.445}_{(0.358, 0.523)}$ |
| **MIMIC-IV** | | | | |
| MMTM (Joze et al., 2020) | $0.855_{(0.840, 0.869)}$ | $0.519_{(0.477, 0.561)}$ | $0.765_{(0.747, 0.783)}$ | $0.465_{(0.430, 0.501)}$ |
| DAFT (Pölsterl et al., 2021) | $0.857_{(0.841, 0.870)}$ | $0.526_{(0.487, 0.565)}$ | $0.765_{(0.747, 0.782)}$ | $0.476_{(0.442, 0.510)}$ |
| Unified (Hayat et al., 2021) | $0.854_{(0.839, 0.870)}$ | $0.505_{(0.545, 0.463)}$ | $0.759_{(0.742, 0.776)}$ | $0.470_{(0.436, 0.503)}$ |
| MedFuse (Hayat et al., 2022) | $0.855_{(0.840, 0.870)}$ | $0.500_{(0.458, 0.541)}$ | $0.762_{(0.744, 0.778)}$ | $0.465_{(0.430, 0.501)}$ |
| DrFuse (Yao et al., 2024) | $0.857_{(0.841, 0.872)}$ | $0.518_{(0.479, 0.562)}$ | $0.768_{(0.749, 0.784)}$ | $0.485_{(0.451, 0.520)}$ |
| CM$^2$ | $\mathbf{0.858}_{(0.844, 0.872)}$ | $\mathbf{0.527}_{(0.490, 0.568)}$ | $\mathbf{0.771}_{(0.752, 0.788)}$ | $\mathbf{0.486}_{(0.452, 0.518)}$ |

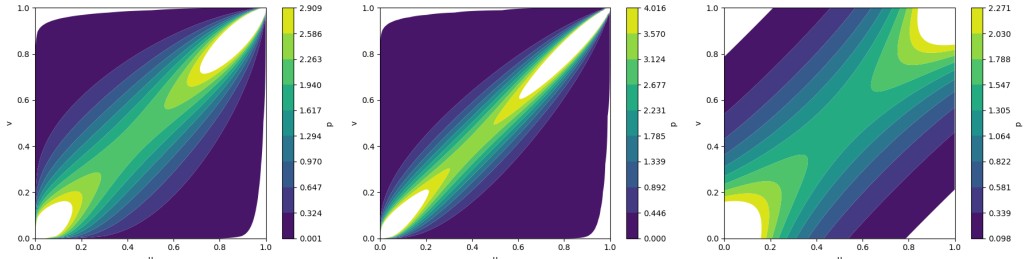

Figure 2: Plots of the fitted copula density to demonstrate the interrelationship captured by the copula model (Left: Gumbel, middle: Gaussian, right: Frank).

MIMIC-III, contributing to the difference in performance between the two datasets under the totally matched setting.

**Qualitative Analysis.** We visualize the copula density of different families of copula and see how the interactions between modalities are captured. Figure 2 presents the visualizations of learned copula densities of the Gumbel, Gaussian, and Frank copula families, respectively. We observe that the Gumbel copula is more focused on the positive dependence between the modalities while the Gaussian copula has fewer weight on modelling tail dependences. On the other hand, the Frank copula is tail-symmetric and capable of modelling both positive and negative dependence. Hence it can cover more dependency structures, indicating that it may be a more flexible choice for modelling complex

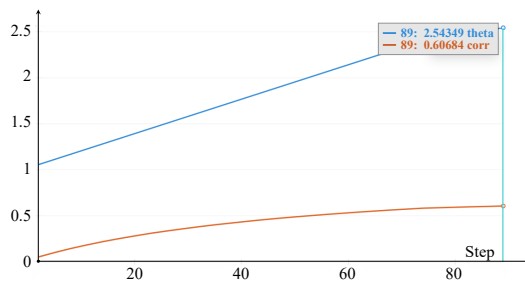

Figure 3: Plots comparing the $\alpha$ value and the correlation (Corr $= \frac{\alpha-1}{\alpha}$) learned by the Gumbel copula model

interactions. We further demonstrate how the CM$^2$ learns the interaction through density plots at different epochs. The detailed discussion can be found in the Appendix. We also study how CM$^2$ learns the correlation over epochs. Figure 3 presents the change in estimated $\alpha$ and its corresponding correlation $\frac{\alpha-1}{\alpha}$ over training epochs. We discover that the model is learning a positive correlation over the epochs, and the correlation converges at around 0.601. This implies that by backpropagating

Table 4: Ablation study on the influence of different components (e.g., resampling and fusion module) of our proposed method. All results are reported in AUROC and AUPR with 95% confidence intervals on MIMIC-IV.

| Model | Matched | IHM | | READM | |
|---|---|---|---|---|---|
| | | AUROC (↑) | AUPR (↑) | AUROC (↑) | AUPR (↑) |
| w/o resampling | × | $0.858_{(0.844,\ 0.872)}$ | $0.521_{(0.485,\ 0.562)}$ | $0.763_{(0.745,\ 0.782)}$ | $0.473_{(0.438,\ 0.511)}$ |
| w/o fusion module | × | $0.860_{(0.845,\ 0.875)}$ | $0.531_{(0.490,\ 0.575)}$ | $0.762_{(0.744,\ 0.781)}$ | $0.476_{(0.442,\ 0.514)}$ |
| w/o fusion module | ✓ | $0.811_{(0.778,\ 0.845)}$ | $0.446_{(0.778,\ 0.845)}$ | $0.720_{(0.685,\ 0.756)}$ | $0.424_{(0.368,\ 0.491)}$ |
| $CM^2$ | × | $0.858_{(0.844,\ 0.872)}$ | $0.527_{(0.490,\ 0.568)}$ | $0.771_{(0.752,\ 0.788)}$ | $0.486_{(0.452,\ 0.518)}$ |
| $CM^2$ | ✓ | $0.827_{(0.790,0.859)}$ | $0.492_{(0.423,0.566)}$ | $0.737_{(0.704,\ 0.773)}$ | $0.466_{(0.404,\ 0.529)}$ |

Table 5: Result on different copula families and the influence of the missing modality. All results are reported in AUROC and AUPR with 95% confidence intervals on MIMIC-IV.

| Model | Matched | Copula Family | IHM | | READM | |
|---|---|---|---|---|---|---|
| | | | AUROC (↑) | AUPR (↑) | AUROC (↑) | AUPR (↑) |
| $CM^2$ | ✓ | Gumbel | $0.825_{(0.792,0.854)}$ | $0.488_{(0.409,0.564)}$ | $0.735_{(0.696,0.772)}$ | $0.463_{(0.405,0.568)}$ |
| $CM^2$ | × | Gumbel | $0.858_{(0.843,\ 0.873)}$ | $0.527_{(0.492,\ 0.568)}$ | $0.772_{(0.789,\ 0.753)}$ | $0.485_{(0.451,\ 0.521)}$ |
| $CM^2$ | ✓ | Frank | $0.827_{(0.790,0.859)}$ | $0.492_{(0.423,0.566)}$ | $0.737_{(0.704,\ 0.773)}$ | $0.466_{(0.404,\ 0.529)}$ |
| $CM^2$ | × | Frank | $0.858_{(0.844,\ 0.872)}$ | $0.527_{(0.490,\ 0.568)}$ | $0.771_{(0.752,\ 0.788)}$ | $0.486_{(0.452,\ 0.518)}$ |
| $CM^2$ | ✓ | Gaussian | $0.827_{(0.791,\ 0.856)}$ | $0.488_{(0.410,\ 0.561)}$ | $0.736_{(0.701,\ 0.772)}$ | $0.458_{(0.528,\ 0.528)}$ |
| $CM^2$ | × | Gaussian | $0.859_{(0.842,\ 0.871)}$ | $0.527_{(0.498,\ 0.560)}$ | $0.771_{(0.754,\ 0.788)}$ | $0.485_{(0.450,\ 0.512)}$ |

the gradient to the copula parameter $\alpha$, the model can learn the interactions between the modalities during training.

## 4.4 ABLATION ANALYSIS

**Effectiveness of Copula Alignment.** We analyze the impact of different alignment loss functions on the performance of $CM^2$. Table 6 presents the results of $CM^2$ on the MIMIC-IV dataset using various align loss functions. Notably, the copula loss consistently outperforms both the cosine loss and KL-divergences (KL) loss, highlighting its effectiveness in modeling the joint distribution of modalities and capturing their interactions.

**Ablation on Contribution of the Designed Modules.** To further evaluate the performance of $CM^2$, we conduct an ablation study by removing the resampling and fusion modules. Table 4 presents the results of $CM^2$ under these modifications. We observe a slight performance decline when the resampling module is removed, indicating its effectiveness in generating unbiased representations for observations with missing modalities. Additionally, the removal of the fusion module results in a significant drop in performance in most cases, highlighting the critical role the fusion module plays in capturing the interactions between modalities and enhancing model performance.

**Ablation on Different Families of Copula.** Beyond the resampling and fusion modules, we also compare the performance of $CM^2$ under different settings for missing modalities and copula families. The accuracy relies heavily on the guesses of the Copula family (Zeng & Wang, 2022). We examine the performance of our method

Table 6: Ablation study on different loss functions. All results are reported in AUROC and AUPR with 95% confidence intervals on MIMIC-IV.

| Align Loss | IHM | | READM | |
|---|---|---|---|---|
| | AUROC (↑) | AUPR (↑) | AUROC (↑) | AUPR (↑) |
| Cosine Loss | $0.820_{(0.784,\ 0.852)}$ | $0.470_{(0.399,\ 0.545)}$ | $0.726_{(0.690,\ 0.762)}$ | $0.445_{(0.387,\ 0.516)}$ |
| KL Loss | $0.826_{(0.792,\ 0.857)}$ | $0.489_{(0.415,\ 0.568)}$ | $0.731_{(0.693,\ 0.766)}$ | $0.446_{(0.391,\ 0.511)}$ |
| Copula Loss | $0.827_{(0.790,0.859)}$ | $0.492_{(0.423,0.566)}$ | $0.737_{(0.704,\ 0.773)}$ | $0.466_{(0.404,\ 0.529)}$ |

over an array of commonly used copula families. Table 11 presents the results of $CM^2$ on MIMIC-IV datasets. We discover that while our method is generally robust to the choice of copula family, the best-performing copula family varies across tasks. This indicates that different tasks highlight different characteristics (e.g. extreme values for mortality) that can be captured when a proper copula family is chosen.

Table 7: Result on MIMIC-IV datasets with three modalities (EHR time series, CXR images, and CXR reports). All results are reported in AUROC and AUPR with 95% confidence intervals.

| Model | IHM | | READM | |
|---|---|---|---|---|
| | AUROC ($\uparrow$) | AUPR ($\uparrow$) | AUROC ($\uparrow$) | AUPR ($\uparrow$) |
| MMTM (Joze et al., 2020) | $0.777_{(0.739, 0.813)}$ | $0.370_{(0.312, 0.443)}$ | $0.689_{(0.650, 0.723)}$ | $0.401_{(0.347, 0.463)}$ |
| DAFT (Pölsterl et al., 2021) | $0.788_{(0.754, 0.821)}$ | $0.397_{(0.331, 0.471)}$ | $0.706_{(0.670, 0.742)}$ | $0.403_{(0.346, 0.464)}$ |
| Unified (Hayat et al., 2021) | $0.795_{(0.761, 0.827)}$ | $0.420_{(0.351, 0.497)}$ | $0.715_{(0.679, 0.749)}$ | $0.430_{(0.376, 0.495)}$ |
| MedFuse (Hayat et al., 2022) | $0.801_{(0.767, 0.836)}$ | $0.427_{(0.367, 0.511)}$ | $0.713_{(0.675, 0.749)}$ | $0.419_{(0.356, 0.487)}$ |
| DrFuse (Yao et al., 2024) | $0.808_{(0.773, 0.839)}$ | $0.451_{(0.376, 0.524)}$ | $0.728_{(0.691, 0.761)}$ | $0.433_{(0.370, 0.495)}$ |
| $CM^2$ | $\mathbf{0.824}_{(0.793, 0.856)}$ | $\mathbf{0.471}_{(0.399, 0.554)}$ | $\mathbf{0.730}_{(0.694, 0.764)}$ | $\mathbf{0.444}_{(0.385, 0.509)}$ |

**Extension to More Modalities.** We further investigate the impact of incorporating more auxiliary modality. We adapt all baselines into the tri-modal setting. Table 7 presents the results for $CM^2$ and the baselines on the MIMIC-IV dataset under tri-modal setting: EHR time series, CXR images, and radiology reports. Across both tasks, $CM^2$ consistently outperforms the baselines, achieving the highest performance. Notably, the baseline models show a decline in performance compared to the bi-modal setting, suggesting that the difficulty of incorporating additional modalities increases as the complexity of aligning them grows with the number of modalities. Despite this, $CM^2$ maintains strong performance, demonstrating its robustness and effectiveness in aligning multiple modalities.

## 5 CONCLUSION

In this work, we introduce copula modelling into multimodal representation learning. Using a copula can effectively model the interaction between the modalities, and impute the missing modalities through sampling from learned marginals. Empirical evaluation validates the predictive performance on multimodal learning tasks, on both the fully and partially matched datasets. Ablation studies showed that the proposed copula model can serve as a promising modality alignment tool and the consistent satisfactory performance over different copula families. Our work can be potentially extended to works that require effective fusion or distribution alignment, including domain adaptation, multi-feature and multi-view learning.

**Limitations and Future Works.** Using a neural network to learn the copula parameter $\alpha$ may be insufficient (since the joint log-likelihood may not be necessarily convex). Hence an alternative updating algorithm (e.g., partial likelihood) is needed in future development of copula multimodal learning to ensure that each loss is convex and we can apply gradient descent. In addition, we select healthcare datasets to demonstrate the effectiveness of our model, while we will extend our method to other multimodal datasets in future works.

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

SUMMARY

In this appendix, we first present detailed information on the datasets in A.1 and tasks used in the experiments in A.2. Next, we introduce the multivariate Gaussian distribution in B and some common copula families in C. Then in D, we discuss the implications of how the copula model learns interactions over the epochs. Finally, we provide more details on the implementation and hyperparameters used in the experiments in E.1 along with the baseline methods settings in E.2.

# A  ADDITIONAL INFORMATION ON DATASETS AND TASKS

## A.1  DATASETS

Table 8 provides a summary of the datasets used in our experiments.

**MIMIC-III dataset** This dataset contains 46,520 ICU stays, each with 17 clinical variables. We split the dataset into training, validation, and test sets in a 70%-15%-15% ratio, following the procedure in (Harutyunyan et al., 2019).

**MIMIC-IV dataset** This dataset includes 21,139 ICU stays, also with 17 clinical variables. The data is split into 70% training, 10% validation, and 20% test sets, following (Hayat et al., 2022).

For both MIMIC-III and MIMIC-IV datasets, we extract 17 clinical variables commonly monitored in the ICU, including 5 categorical and 12 continuous variables. Data are sampled every two hours during the first 48 hours of ICU admission for both tasks, in accordance with (Hayat et al., 2022). This results in a vector representation of size 76 at each time step of the clinical time-series data.

**MIMIC-CXR dataset** This dataset contains 377,110 chest X-ray images, of which 5,931 are associated with MIMIC-IV ICU stays. We split the data into 4,287 training samples, 465 validation samples, and 1,179 test samples. Following (Hayat et al., 2022), we retrieve the last Anterior-Posterior (PA) projection chest X-ray and apply transformations to the images, resizing them to $224 \times 224$ pixels.

This dataset also includes radiology reports, which are unstructured text data. We choose the radiology reports of the MIMIC-CXR dataset as an auxiliary modality to investigate the effectiveness of $CM^2$ on more modality alignment since the radiology reports do not contain death information and can avoid possible overfitting and shortcuts. We divide the unstructured radiology reports into 4 sections, including Impression, Findings, Last paragraph, and Comparison.

**MIMIC-III NOTE dataset** This dataset consists of 5,273 clinical notes associated with MIMIC-III ICU stays. The data is divided into 3,652 training samples, 815 validation samples, and 806 test samples. In line with (Zhang et al., 2023), we select the last five clinical notes before the prediction time. If fewer than five notes are available, we treat the notes for that ICU stay as missing. The original number of matched ICU stays is around 15,000. We randomly sample one-third of the matched ICU stays to form the training, validation, and test sets, keeping the scale of the notes nearly the same as the CXRs in the MIMIC-IV dataset.

Both radiology reports sections and clinical notes are capped at a maximum length of 512 words, tokenized into words, and embedded into 312-dimensional vectors using the pre-trained TinyBERT model (Jiao et al., 2019)[1].

## A.2  TASKS

**In Hospital Mortality (IHM) Prediction.** The In Hospital Mortality (IHM) prediction task focuses on predicting whether a patient will pass away during their hospital stay. As summarized in Table 8, the MIMIC-III dataset contains a total of 2,795 positive samples, of which 736 are matched with clinical notes. Similarly, the MIMIC-IV dataset includes 3,153 positive samples, with 890 matched to CXR.

---

[1] https://huggingface.co/huawei-noah/TinyBERT_General_4L_312D

Table 8: Datasets Summary

| Dataset | Tasks | No. Train | No. Valid | No. Test | No. Pos. | Total |
|---|---|---|---|---|---|---|
| | | *Complete Datasets* | | | | |
| **MIMIC-III** | IHM | 14681 | 3222 | 3236 | 2795 | 21139 |
| **MIMIC-III** | READM | 14681 | 3222 | 3236 | 3987 | 21139 |
| **MIMIC-III NOTE** | - | 3652 | 815 | 806 | - | 5,273 |
| **MIMIC-IV** | IHM | 18064 | 2035 | 4972 | 3153 | 25071 |
| **MIMIC-IV** | READM | 18064 | 2035 | 4972 | 4603 | 25071 |
| **MIMIC-CXR** | - | 344529 | 9497 | 23069 | - | 377,095 |
| | | *Matched Datasets* | | | | |
| **MIMIC-III | NOTE** | IHM | 3652 | 815 | 806 | 736 | 5273 |
| **MIMIC-III | NOTE** | READM | 3652 | 815 | 806 | 998 | 5273 |
| **MIMIC-IV | CXR** | IHM | 4287 | 465 | 1179 | 890 | 5931 |
| **MIMIC-IV | CXR** | READM | 4287 | 465 | 1179 | 1262 | 5931 |
| **MIMIC-IV | CXR | REPORT** | IHM | 4287 | 465 | 1179 | 890 | 5931 |
| **MIMIC-IV | CXR | REPORT** | READM | 4287 | 465 | 1179 | 1262 | 5931 |

**Readmission (READM) Prediction.** The Readmission (READM) prediction task aims to forecast whether a patient will be readmitted within 30 days of discharge. In this task, both patients who are readmitted and those who pass away in hospital are considered positive samples. As shown in Table 8, the MIMIC-III dataset contains 3,987 positive samples, with 998 matched to clinical notes. In the MIMIC-IV dataset, there are 4,603 positive samples, with 1,262 matched to CXRs.

## B  MULTIVARIATE GAUSSIAN DISTRIBUTION

The multivariate Gaussian Distribution is defined as

$$p(\boldsymbol{z}; \boldsymbol{\mu}, \boldsymbol{\Sigma}) = \frac{1}{(2\pi)^{\frac{n}{2}} |\boldsymbol{\Sigma}|^{\frac{1}{2}}} \exp\left\{ -\frac{1}{2}(\boldsymbol{z} - \boldsymbol{\mu})^T \boldsymbol{\Sigma}^{-1} (\boldsymbol{z} - \boldsymbol{\mu}) \right\}$$

where $\boldsymbol{\mu} \in \mathbb{R}^p$ is a $p$-dimensional mean vector and $\boldsymbol{\Sigma} \in \mathbb{R}^{p \times p}$ is the covariance matrix.

The KL divergences of two multivariate normal distributions $\mathcal{N}(\boldsymbol{\mu}_1, \Sigma_1)$ and $\mathcal{N}(\boldsymbol{\mu}_2, \Sigma_2)$

$$KL(\mathcal{N}(\boldsymbol{\mu}_1, \boldsymbol{\Sigma}_1) \| \mathcal{N}(\boldsymbol{\mu}_2, \boldsymbol{\Sigma}_2)) =$$
$$\frac{1}{2}\left[ \log \frac{|\boldsymbol{\Sigma}_2|}{|\boldsymbol{\Sigma}_1|} - p + \mathrm{tr}\{\boldsymbol{\Sigma}_2^{-1}\boldsymbol{\Sigma}_1\} + (\boldsymbol{\mu}_2 - \boldsymbol{\mu}_1)^T \boldsymbol{\Sigma}_2^{-1}(\boldsymbol{\mu}_2 - \boldsymbol{\mu}_1) \right]$$

## C  COMMON COPULA FAMILIES.

We specify the copula distribution and density functions of common copula families with necessary derivations. We consider bivariate copula without loss of generality.

**Archimedean Copula.** A subclass of copulas that can be constructed easily by the use of generator functions $\varphi : [0,1] \rightarrow [0, \infty]$, which are strictly decreasing and convex so that $\varphi(0) = \infty$ and $\varphi(1) = 0$. Then, a copula $C$ can be constructed:

$$C(u_1, u_2, \ldots, u_d) = \varphi^{[-1]}\left( \sum_{i=1}^{d} \varphi(u_i) \right).$$

The Archimedean copula can generate copula densities when there is more than one modality in the dataset.

## C.1 COPULA DISTRIBUTION FUNCTIONS

- Clayton

$$C(u, v; \alpha) = \left[ \max\{ u^{-\alpha} + v^{-\alpha} - 1, 0 \} \right]^{-1/\alpha}$$

- Frank

$$C(u, v; \alpha) = -\frac{1}{\alpha} \log \left[ 1 - \frac{(1 - e^{\alpha u})(1 - e^{\alpha v}))}{1 - e^{-\alpha}} \right],$$

where $\alpha \in \mathbb{R} \backslash \{0\}$.

- Gumbel

$$C(u, v; \alpha) = \exp\{ - [(- \log u)^{\alpha} + (- \log v)^{\alpha}]^{\frac{1}{\alpha}} \}$$

- Gaussian

$$C(u, v; \rho) = \Phi_2 \left[ \Phi^{-1}(u), \Phi^{-1}(v), \rho \right],$$

where $\Phi$ is the cdf of a standard Gaussian distribution.

- Student's $t$

$$C(u, v; \rho, \nu) = T_{2,\nu}[T_\nu^{-1}(u), T_\nu^{-1}(v); \rho], \qquad v > 0; |\rho| < 1,$$

## C.2 COPULA DENSITY FUNCTIONS

**Clayton Copula**

$$c(u, v) = (1 + \alpha)(uv)^{-1-\alpha}(-1 + u^{-\alpha} + v^{-\alpha})^{-2-1/\alpha},$$

where $\alpha \in (-1, \infty)$

**Frank Copula**

$$c(u, v) = \frac{-\alpha e^{-\alpha(u+v)}(e^{-\alpha} - 1)}{(e^{-\alpha} - e^{-\alpha u} - e^{-\alpha v} + e^{-\alpha(u+v)})^2}$$

where $\alpha \in (-\infty, \infty), \alpha \neq 0$.

**Gumbel copula**

$$c(u,v) = \frac{\partial}{\partial u}\frac{\partial}{\partial v}C(u,v)$$

$$= \frac{\partial}{\partial u}\frac{\partial}{\partial v}\exp\{-[(-\log u)^\alpha + (-\log v)^\alpha]^{1/\alpha}\}$$

$$:= \frac{\partial}{\partial u}\frac{\partial}{\partial v}\exp\{-[g(u,v;\alpha)]^{1/\alpha}\}$$

$$= \frac{\partial}{\partial u} - \exp\{-[g(u,v;\alpha)]^{1/\alpha}\}\left(\frac{1}{\alpha}[g(u,v;\alpha)]^{\frac{1-\alpha}{\alpha}}\right)\frac{\partial}{\partial v}g(u,v;\alpha)$$

$$= \frac{\partial}{\partial u}\exp\{-[g(u,v;\alpha)]^{1/\alpha}\}\left(\frac{1}{\alpha}[g(u,v;\alpha)]^{\frac{1-\alpha}{\alpha}}\right)\alpha(-\log v)^{\alpha-1}\frac{1}{v}$$

$$= \frac{\alpha}{v}(-\log v)^{\alpha-1}\left[\left(\frac{1}{\alpha}[g(u,v;\alpha)]^{\frac{1-\alpha}{\alpha}}\right)\frac{\partial}{\partial u}\exp\{-[g(u,v;\alpha)]^{1/\alpha}\}\right.$$

$$\left.+ \exp\{-[g(u,v;\alpha)]^{1/\alpha}\}\frac{\partial}{\partial u}\frac{1}{\alpha}[g(u,v;\alpha)]^{\frac{1-\alpha}{\alpha}}\right]$$

$$= \frac{\alpha}{v}(-\log v)^{\alpha-1}\left[-\frac{1}{\alpha}[g(u,v;\alpha)]^{\frac{1-\alpha}{\alpha}}\exp\{-[g(u,v;\alpha)]^{1/\alpha}\}\left(\frac{1}{\alpha}[g(u,v;\alpha)]^{\frac{1-\alpha}{\alpha}}\right)\frac{\partial}{\partial u}g(u,v;\alpha)\right.$$

$$\left.+ \exp\{-[g(u,v;\alpha)]^{1/\alpha}\}\frac{1-\alpha}{\alpha^2}[g(u,v;\alpha)]^{\frac{1-2\alpha}{\alpha}}\frac{\partial}{\partial u}g(u,v;\alpha)\right]$$

$$= \frac{\alpha}{v}(-\log v)^{\alpha-1}\frac{\partial}{\partial u}g(u,v;\alpha)\exp\{-[g(u,v;\alpha)]^{1/\alpha}\}\left[\frac{1}{\alpha^2}[g(u,v;\alpha)]^{\frac{2(1-\alpha)}{\alpha}}\right.$$

$$\left.+ \frac{\alpha-1}{\alpha^2}[g(u,v;\alpha)]^{\frac{1-2\alpha}{\alpha}}\right]$$

$$= \frac{1}{uv}(-\log v)^{\alpha-1}(-\log u)^{\alpha-1}C(u,v)\left[(\alpha-1)[g(u,v;\alpha)]^{\frac{1-2\alpha}{\alpha}} + [g(u,v;\alpha)]^{\frac{2(1-\alpha)}{\alpha}}\right]$$

$$= \frac{1}{uv}(-\log v)^{\alpha-1}(-\log u)^{\alpha-1}C(u,v)[g(u,v;\alpha)]^{\frac{2(1-\alpha)}{\alpha}}\left[(\alpha-1)[g(u,v;\alpha)]^{-\frac{1}{\alpha}} + 1\right]$$

The closed-form density of the trivariate Gumbel Copula is computed by

$$c(u,v,w) = \frac{\partial}{\partial u}\frac{\partial}{\partial v}\frac{\partial}{\partial w}C(u,v,w)$$

$$= \frac{\partial}{\partial u}\frac{\partial}{\partial v}\frac{\partial}{\partial w}\exp\{-[(-\log u)^\alpha + (-\log v)^\alpha + (-\log w)^\alpha]^{1/\alpha}\}$$

$$:= \frac{\partial}{\partial u}\frac{\partial}{\partial v}\frac{\partial}{\partial w}\exp\{-(h(u,v,w;\alpha))^\alpha\}$$

$$= \frac{1}{uvw}(-\log(u))^{\alpha-1}(-\log(v))^{\alpha-1}(-\log(w))^{\alpha-1}C(u,v,w)$$

$$\cdot \left(\alpha^6(h(u,v,w;\alpha))^{3\alpha-3} - (\alpha-1)\alpha^5(h(u,v,w;\alpha))^{2\alpha-3} - 2\alpha^5(\alpha-1)(h(u,v,w;\alpha))^{2\alpha-3}\right.$$

$$\left.+ (\alpha-2)(\alpha-1)\alpha^4(h(u,v,w;\alpha))^{\alpha-3}\right)$$

The identity can be generated by the Archimedean copula for $M > 3$, which is less common in multimodal learning:

$$c(\boldsymbol{u}) = \psi^{(d)}(t(\boldsymbol{u}))\prod_{j=1}^{d}(\psi^{-1})'(u_j),$$

where $\varphi(t;\alpha) = (\log t)^\alpha$ for the Gumbel copula.

**Gaussian Copula** The bivariate case is given by

$$c(u,v;\rho) = \frac{1}{\sqrt{1-\rho}}\exp\left(-\frac{(a^2+b^2)\rho^2 - 2ab\rho}{2(1-\rho^2)}\right),$$

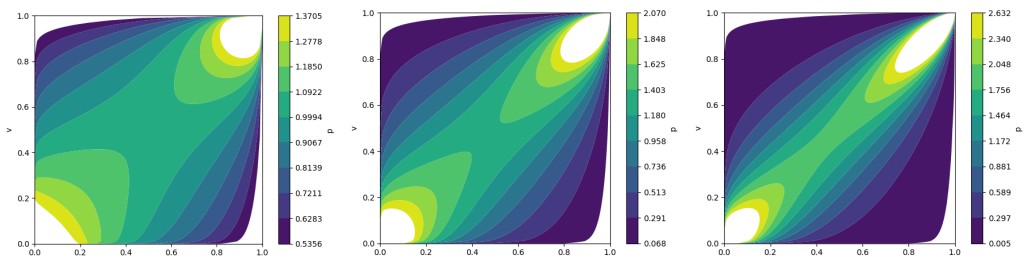

Figure 4: Plots of the copula densities of the Gumbel family at epochs 5, 50, and 100, respectively.

where $a = \sqrt{2}\text{erf}^{-1}(2u - 1)$, and $b = \sqrt{2}\text{erf}^{-1}(2v - 1)$. The multivariate case is given by the following matrix form

$$c(u, v; \rho)$$

**Student's $t$ Copula**

$$\frac{\Gamma(0.5v)\Gamma(0.5v + 1)(1 + (t_v^{-2}(u) + t_v^{-2}(v) - 2\rho t_v^{-1}(u)t_v^{-1}(v))/(v(1 - \rho^2))^{-0.5(v+2)})}{\sqrt{1 - \rho^2}\Gamma(0.5(v + 1))^2(1 + t_v^{-2}(u)/v)^{-0.5(v+1)}(1 + t_v^{-2}(v)/v)^{-0.5(v+1)}},$$

where $v$ is the degree of freedom, $\Gamma$ is the gamma function, $t_v(x; v) = \int_{-\infty}^{x} \frac{\Gamma(0.5(v + 1))}{\sqrt{v\pi}\Gamma(0.5v)(1 + v^{-1}t^2)^{0.5(v+1)}}$.

# D    DISCUSSION ON HOW COPULA MODEL LEARNS INTERACTIONS.

We demonstrate how the copula model learns the interactions over the epochs and further discuss the implications.

Figure 4 presents the copula densities at epochs epochs 5, 50, and 100, respectively. We use the Gumbel family as an illustrative example. We observe that the copula density is evolving to a positive correlation pattern, while the negative correlation scenarios (e.g., $u > 0.5, v < 0.5$, or $u < 0.5, v > 0.5$) are still considered but the weights allocated are decreasing.

# E    MORE ON BASELINE METHODS AND IMPLEMENTATION DETAILS

## E.1    IMPLEMENTATION DETAILS AND HYPERPARAMETERS

We train all models for 100 epochs on the training set and select the best-performing model based on the validation set, using the AUROC as the monitoring metric. The final results are reported on the test set. We optimize the models using the *Adam* optimizer and apply early stopping if the validation AUROC does not improve for 15 consecutive epochs to prevent overfitting. All experiments are conducted on a single RTX-3090 GPU. The batch size is set to 32 for models trained on the MIMIC-IV & CXR datasets, and 16 for models trained on the MIMIC-III & NOTE datasets, except for DrFuse, which is trained with a batch size of 8. We employ grid search to tune hyperparameters using the validation set and report the best results on the test set. The hyperparameter search space includes:

- Dropout ratio: $\{0, 0.1, 0.2, 0.3\}$
- Learning rate: $\{1 \times 10^{-4}, 5 \times 10^{-5}, 1 \times 10^{-5}\}$
- Number of Gaussian mixture K: $\{2, 3, 4, 5, 6\}$
- Temperature: $\{0.001, 0.005, 0.01, 0.05, 0.08\}$
- Regularization parameter $\lambda_{\text{cop}}$: $\{1 \times 10^{-5}, 5 \times 10^{-6}, 1 \times 10^{-6}\}$

CM$^2$ is implemented in Python 3.11 using *PyTorch* 1.9. Following MedFuse (Hayat et al., 2022), we use ResNet34 (He et al., 2016) as the backbone encoder for CXR, a two-layer LSTM (Graves & Graves, 2012) as the encoder for time-series data, and pre-trained TinyBERT (Jiao et al., 2019)[2] as the encoder for clinical notes. We include a projection layer to map modality embeddings into the same latent space. A two-layer LSTM is used as the fusion module to combine modality embeddings, and a multilayer perceptron (MLP) with one linear layer and a sigmoid activation function serves as the classifier.

### E.2 ADDITIONAL SETTINGS OF BASELINE METHODS

We compare CM$^2$ with the following baseline methods.

- **MMTM** (Joze et al., 2020) is a module that can leverage the information between modalities with flexible plugin architectures. Since the model assumes full modality, we compensate for the missing modality CXR and clinical notes with all zeros during training and testing. For clinical notes, we replace the ResNet34 encoder with TinyBERT to embed the clinical notes.

- **DAFT** (Pölsterl et al., 2021) is a module that can be plugged into CNN models to exchange information between tabular data and image modality. Similarly, we replace the input of CXR and clinical notes with matrices of all zeros during training and testing and use TinyBERT to embed the clinical notes.

- **Unified** (Hayat et al., 2021) is a dynamic approach towards integrating auxiliary data modalities, learning the data representations for the individual modalities, and integrating the representations via a unified classifier. It inherently handles missingness and leverages all of the available modality-specific data. Also, we use TinyBERT to embed the clinical notes.

- **MedFuse** (Hayat et al., 2022) uses an LSTM-based fusion to combine features from the image encoder (or language encoder) and EHR encoder. Missing modality is handled by learning a global representation for the missing CXR or clinical notes. We randomly initialized encoders for the time-series data, clinical notes, and CXR images.

- **DrFuse** (Yao et al., 2024) uses disentangled representation learning to learn a shared representation between the EHR and image modality even when one modality is missing. Drfuse uses ResNet50 as the image encoder and Transformer as the EHR encoder. We replace the ResNet50 encoder with TinyBERT to embed the clinical notes.

The Implementation of DrFuse follows the original paper(Yao et al., 2024)[3], and we use the same hyperparameters as the original paper. We directly adopt the implementations of MMTM, DAFT, Unified, and MedFuse provided by (Hayat et al., 2022)[4], and all hyperparameters are set to the default values provided by Hayat et al. (2022). We adapt the implementations of MMTM, DAFT, Unified, MedFuse and DrFuse to tri-modal setting, including EHR time-series data, CXR images, and radiology reports.

## F ADDITIONAL EXPERIMENT RESULTS

**Additional Baselines.** We add two more baselines, LSMT (Khader et al., 2023) and Interleaved (Zhang et al., 2023), to compare with CM$^2$. The results are shown in Table 9.

- **LSMT** (Khader et al., 2023) is a transformer-based model designed for the multimodal medical context.

- **Interleaved** (Zhang et al., 2023) is a multimodal approach that addresses the irregularity of medical multimodal data and fuses representations from different modalities using cross-modal attention.

---

[2] https://huggingface.co/huawei-noah/TinyBERT_General_4L_312D
[3] https://github.com/dorothy-yao/drfuse
[4] https://github.com/nyuad-cai/MedFuse

Table 9: Results of Additional Baselines on MIMIC-IV datasets. All results are reported in AUROC and AUPR with 95% confidence intervals. The best results are highlighted in **bold.**

| Model | IHM | | READM | |
|---|---|---|---|---|
| | AUROC (↑) | AUPR (↑) | AUROC (↑) | AUPR (↑) |
| **Totally Matched** | | | | |
| LSMT (Khader et al., 2023) | $0.803_{(0.769, 0.837)}$ | $0.444_{(0.370, 0.519)}$ | $0.701_{(0.662, 0.737)}$ | $0.421_{(0.356, 0.490)}$ |
| Interleaved (Zhang et al., 2023) | $0.800_{(0.764, 0.834)}$ | $0.440_{(0.374, 0.523)}$ | $0.702_{(0.664, 0.741)}$ | $0.421_{(0.360, 0.487)}$ |
| $\text{CM}^2$ | $\mathbf{0.827}_{(0.790, 0.859)}$ | $\mathbf{0.492}_{(0.423, 0.566)}$ | $\mathbf{0.737}_{(0.704, 0.773)}$ | $\mathbf{0.466}_{(0.404, 0.529)}$ |
| **Partially Matched-IV** | | | | |
| LSMT (Khader et al., 2023) | $0.854_{(0.838, 0.870)}$ | $0.508_{(0.466, 0.551)}$ | $0.764_{(0.746, 0.781)}$ | $0.473_{(0.436, 0.509)}$ |
| Interleaved (Zhang et al., 2023) | $0.856_{(0.840, 0.871)}$ | $0.508_{(0.466, 0.550)}$ | $0.758_{(0.740, 0.775)}$ | $0.473_{(0.441, 0.506)}$ |
| $\text{CM}^2$ | $\mathbf{0.858}_{(0.844, 0.872)}$ | $\mathbf{0.527}_{(0.490, 0.568)}$ | $\mathbf{0.771}_{(0.752, 0.788)}$ | $\mathbf{0.486}_{(0.452, 0.518)}$ |

Table 10: Results of different backbone encoders and additional baselines on MIMIC-IV datasets with *totally matched* modalities. All results are reported in AUROC and AUPR with 95% confidence intervals. The best results are highlighted in **bold**.

| Model | Backbone TS | Backbone IMG | IHM AUROC (↑) | IHM AUPR (↑) | READM AUROC (↑) | READM AUPR (↑) |
|---|---|---|---|---|---|---|
| MMTM (Joze et al., 2020) | LSTM | ResNet | $0.802_{(0.770, 0.835)}$ | $0.429_{(0.362, 0.513)}$ | $0.713_{(0.677, 0.750)}$ | $0.420_{(0.362, 0.489)}$ |
| DAFT (Pölsterl et al., 2021) | | | $0.815_{(0.782, 0.844)}$ | $0.454_{(0.387, 0.538)}$ | $0.729_{(0.692, 0.766)}$ | $0.433_{(0.378, 0.499)}$ |
| Unified (Hayat et al., 2021) | | | $0.808_{(0.778, 0.840)}$ | $0.429_{(0.367, 0.512)}$ | $0.719_{(0.680, 0.756)}$ | $0.450_{(0.390, 0.513)}$ |
| MedFuse (Hayat et al., 2022) | | | $0.813_{(0.777, 0.844)}$ | $0.448_{(0.380, 0.528)}$ | $0.725_{(0.690, 0.762)}$ | $0.438_{(0.379, 0.508)}$ |
| DrFuse (Yao et al., 2024) | | | $0.814_{(0.780, 0.844)}$ | $0.450_{(0.384, 0.536)}$ | $0.723_{(0.687, 0.756)}$ | $0.422_{(0.367, 0.486)}$ |
| LSMT (Khader et al., 2023) | | | $0.803_{(0.769, 0.837)}$ | $0.444_{(0.374, 0.523)}$ | $0.701_{(0.662, 0.737)}$ | $0.421_{(0.356, 0.490)}$ |
| Interleaved (Zhang et al., 2023) | | | $0.800_{(0.764, 0.834)}$ | $0.440_{(0.370, 0.519)}$ | $0.702_{(0.664, 0.741)}$ | $0.421_{(0.360, 0.487)}$ |
| $\text{CM}^2$ | | | $\mathbf{0.827}_{(0.790, 0.859)}$ | $\mathbf{0.492}_{(0.423, 0.566)}$ | $\mathbf{0.737}_{(0.704, 0.773)}$ | $\mathbf{0.466}_{(0.404, 0.529)}$ |
| MMTM (Joze et al., 2020) | LSTM | ViT | $0.805_{(0.768, 0.837)}$ | $0.446_{(0.377, 0.524)}$ | $0.712_{(0.676, 0.749)}$ | $0.422_{(0.360, 0.491)}$ |
| DAFT (Pölsterl et al., 2021) | | | $0.808_{(0.775, 0.840)}$ | $0.438_{(0.365, 0.521)}$ | $0.714_{(0.678, 0.753)}$ | $0.423_{(0.369, 0.490)}$ |
| Unified (Hayat et al., 2021) | | | $0.803_{(0.768, 0.835)}$ | $0.431_{(0.365, 0.515)}$ | $0.707_{(0.667, 0.743)}$ | $0.416_{(0.360, 0.482)}$ |
| MedFuse (Hayat et al., 2022) | | | $0.805_{(0.771, 0.837)}$ | $0.439_{(0.371, 0.524)}$ | $0.715_{(0.677, 0.753)}$ | $0.424_{(0.370, 0.492)}$ |
| DrFuse (Yao et al., 2024) | | | $0.806_{(0.772, 0.838)}$ | $0.446_{(0.379, 0.526)}$ | $0.716_{(0.677, 0.748)}$ | $0.421_{(0.364, 0.489)}$ |
| LSMT (Khader et al., 2023) | | | $0.801_{(0.767, 0.836)}$ | $0.441_{(0.374, 0.527)}$ | $0.703_{(0.662, 0.739)}$ | $0.410_{(0.358, 0.475)}$ |
| Interleaved (Zhang et al., 2023) | | | $0.802_{(0.766, 0.833)}$ | $0.434_{(0.364, 0.509)}$ | $0.710_{(0.673, 0.747)}$ | $0.435_{(0.372, 0.502)}$ |
| $\text{CM}^2$ | | | $\mathbf{0.826}_{(0.790, 0.856)}$ | $\mathbf{0.490}_{(0.421, 0.563)}$ | $\mathbf{0.736}_{(0.697, 0.771)}$ | $\mathbf{0.452}_{(0.394, 0.522)}$ |
| MMTM (Joze et al., 2020) | Transformer | ResNet | $0.813_{(0.780, 0.846)}$ | $0.452_{(0.383, 0.540)}$ | $0.735_{(0.699, 0.770)}$ | $0.448_{(0.388, 0.515)}$ |
| DAFT (Pölsterl et al., 2021) | | | $0.814_{(0.782, 0.845)}$ | $0.437_{(0.373, 0.522)}$ | $0.730_{(0.694, 0.766)}$ | $0.430_{(0.372, 0.493)}$ |
| Unified (Hayat et al., 2021) | | | $0.812_{(0.776, 0.845)}$ | $0.453_{(0.385, 0.533)}$ | $0.719_{(0.681, 0.754)}$ | $0.426_{(0.365, 0.488)}$ |
| MedFuse (Hayat et al., 2022) | | | $0.815_{(0.782, 0.846)}$ | $0.441_{(0.373, 0.520)}$ | $0.728_{(0.692, 0.762)}$ | $0.442_{(0.381, 0.505)}$ |
| DrFuse (Yao et al., 2024) | | | $0.818_{(0.784, 0.850)}$ | $0.460_{(0.391, 0.540)}$ | $0.726_{(0.689, 0.760)}$ | $0.430_{(0.370, 0.495)}$ |
| LSMT (Khader et al., 2023) | | | $0.817_{(0.785, 0.848)}$ | $0.452_{(0.386, 0.535)}$ | $0.722_{(0.688, 0.758)}$ | $0.431_{(0.376, 0.494)}$ |
| Interleaved (Zhang et al., 2023) | | | $0.821_{(0.791, 0.851)}$ | $0.459_{(0.389, 0.539)}$ | $0.721_{(0.683, 0.757)}$ | $0.429_{(0.367, 0.497)}$ |
| $\text{CM}^2$ | | | $\mathbf{0.823}_{(0.788, 0.855)}$ | $\mathbf{0.488}_{(0.421, 0.560)}$ | $\mathbf{0.740}_{(0.699, 0.771)}$ | $\mathbf{0.470}_{(0.382, 0.510)}$ |
| MMTM (Joze et al., 2020) | Transformer | ViT | $0.813_{(0.778, 0.846)}$ | $0.462_{(0.396, 0.545)}$ | $0.723_{(0.686, 0.761)}$ | $0.435_{(0.380, 0.505)}$ |
| DAFT (Pölsterl et al., 2021) | | | $0.803_{(0.768, 0.836)}$ | $0.432_{(0.363, 0.510)}$ | $0.719_{(0.682, 0.758)}$ | $0.421_{(0.367, 0.486)}$ |
| Unified (Hayat et al., 2021) | | | $0.812_{(0.778, 0.845)}$ | $0.463_{(0.396, 0.546)}$ | $0.719_{(0.680, 0.753)}$ | $0.412_{(0.353, 0.474)}$ |
| MedFuse (Hayat et al., 2022) | | | $0.818_{(0.786, 0.849)}$ | $0.461_{(0.393, 0.542)}$ | $0.721_{(0.684, 0.759)}$ | $0.431_{(0.371, 0.493)}$ |
| DrFuse (Yao et al., 2024) | | | $0.814_{(0.780, 0.845)}$ | $0.436_{(0.369, 0.516)}$ | $0.717_{(0.680, 0.755)}$ | $0.416_{(0.359, 0.480)}$ |
| LSMT (Khader et al., 2023) | | | $0.815_{(0.784, 0.847)}$ | $0.453_{(0.389, 0.535)}$ | $0.714_{(0.675, 0.751)}$ | $0.424_{(0.365, 0.492)}$ |
| Interleaved (Zhang et al., 2023) | | | $0.818_{(0.786, 0.849)}$ | $0.453_{(0.380, 0.531)}$ | $0.717_{(0.679, 0.753)}$ | $0.433_{(0.371, 0.498)}$ |
| $\text{CM}^2$ | | | $\mathbf{0.826}_{(0.790, 0.855)}$ | $\mathbf{0.489}_{(0.422, 0.560)}$ | $\mathbf{0.737}_{(0.700, 0.772)}$ | $\mathbf{0.465}_{(0.394, 0.517)}$ |

**Effect of Backbone Encoders.** Moreover, wo explore the effectiveness of backbone encoders for both time-series data and CXR images data. The results are shown in Table 10.

**Effect of Number of Mixture $K$.** The performance of $\text{CM}^2$ with respect to different values of $K$ is shown in Figure 5.

**Statistical Tests** The p-values of two-sample bootstrapped $t$-test of the AUROC and AUPR of $\text{CM}^2$ compared to baseline methods are shown in Table 11.

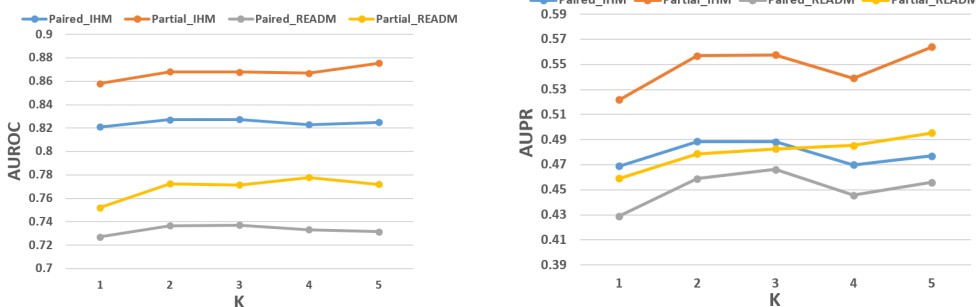

Figure 5: Results (left: AUROC; right: AUPR) of $CM^2$ on MIMIC-IV datasets, where the model reduced to a multivariate Gaussian assumption when $K = 1$.

Table 11: P-values of two-sample bootstrapped $t$-test of the AUROC and AUPR of $CM^2$ compared to baseline methods. We observe that most of the tests are significant under 0.05 significance level.

| Model | IHM | | READM | |
|---|---|---|---|---|
| | AUROC ($\uparrow$) | AUPR ($\uparrow$) | AUROC ($\uparrow$) | AUPR ($\uparrow$) |
| MMTM (Joze et al., 2020) | 2.02e-06 | 3.55e-180 | 4.40e-100 | 5.36e-291 |
| DAFT (Pölsterl et al., 2021) | 0.1122 | 1.53e-132 | 9.37e-78 | 2.95e-240 |
| Unified (Hayat et al., 2021) | 4.55e-08 | 5.71e-240 | 4.80e-73 | 2.81e-139 |
| MedFuse (Hayat et al., 2022) | 7.73e-07 | 5.66e-129 | 1.11e-92 | 3.69e-173 |
| DrFuse (Yao et al., 2024) | 0.1447 | 4.28e-99 | 6.05e-67 | 6.25e-250 |