# OpenReview forum: "Cross-Modal Alignment via Variational Copula Modelling"
_ICLR.cc/2025/Conference — ICLR 2025 Conference Withdrawn Submission_

### Official Review · Reviewer_Zvwa · 2024-10-31

**Soundness:** 2
**Presentation:** 3
**Contribution:** 2
**Rating:** 3
**Confidence:** 4

**Summary:**

In this paper, they propose a copula-driven multimodal learning framework that captures complex interactions by learning the joint distribution of multiple modalities. By aligning marginal distributions with a copula model and assuming a Gaussian mixture distribution for each modality, their approach effectively generates accurate representations for missing modalities. Extensive experiments on public MIMIC datasets demonstrate the model's superior performance over competitors. Ablation studies validate the effectiveness of the copula alignment strategy and the model's robustness across different copula families.

**Strengths:**

There are three major contributions.

First, they propose a copula-driven multimodal learning framework that captures complex interactions by learning the joint distribution of multiple modalities.

Second, they approximate   marginal distributions by assuming a Gaussian mixture distribution for each modality, generating accurate representations for missing modalities.

Third, extensive experiments on public MIMIC datasets demonstrate the model's superior performance over competitors.

**Weaknesses:**

There are several major weaknesses in the paper:

\begin{itemize}
    \item \textbf{First}, if the goal is to integrate multiple modalities to improve prediction, accurately modeling the joint distribution of these modalities may not be necessary.

    \item \textbf{Second}, it is unclear how a Gaussian mixture distribution for each modality generates accurate representations for missing modalities, especially since the authors have not modeled the missing data mechanism.

    \item \textbf{Third}, there are numerous typos and missing details throughout the paper that require correction. For instance, what is \( c_M \) on line 160? What does \( q(\theta) \) represent on line 224? Additionally, the connection between \( f_\theta(x) \) and the copula distribution is not clarified. These omissions make it difficult for readers to understand the technical aspects of the paper.
\end{itemize}

**Questions:**

\begin{itemize}
    \item \textbf{First}, if the goal is to integrate multiple modalities to improve prediction, accurately modeling the joint distribution of these modalities may not be necessary.

    \item \textbf{Second}, it is unclear how a Gaussian mixture distribution for each modality generates accurate representations for missing modalities, especially since the authors have not modeled the missing data mechanism.

    \item \textbf{Third}, there are numerous typos and missing details throughout the paper that require correction. For instance, what is \( c_M \) on line 160? What does \( q(\theta) \) represent on line 224? Additionally, the connection between \( f_\theta(x) \) and the copula distribution is not clarified. These omissions make it difficult for readers to understand the technical aspects of the paper.
\end{itemize}

---

> ### Author Response · Authors · 2024-11-21
> **Thank you for your comments on our manuscripts.**
>
> Thank you for your comments on our manuscripts. We will respond to your comments as follows:
> >>if the goal is to integrate multiple modalities to improve prediction, accurately modeling the joint distribution of these modalities may not be necessary
>
> Thank you for the comments.
> We understand that previous multimodal learning methods focus on the empirical alignment of representations (e.g., contrastive learning via similarity) and the notion of joint distribution is rarely mentioned.
> However, learning the joint distribution is a more advanced way of alignment.
> These ways can inherently be viewed as learning the joint distribution, since their goals are all learning the fused representations (which is equivalent to the joint distribution).
> Moreover, due to the limit nature of the tail dependence coefficients, it is difficult to empirically estimate the tail dependence based on observed data (i.e., tail observations are rare), prompting the need for probabilistic methods.
> As acknowledged by reviewer haPi, using a copula to capture the joint distribution is a simple yet effective way of modelling the cross-modal interactions.
> Hence, we formulate the multimodal learning problem as the problem of estimating the joint distribution, where by placing a statistical model (e.g., copula), we can model the interaction structures between the modalities and obtain a better alignment result.
> Furthermore, learning the joint distribution enables us to generate representations for missing modalities, where the learned distribution contains information on the interactions between modalities modelled by copulas.
>
> >>It is unclear how a Gaussian mixture distribution for each modality generates accurate representations for missing modalities, especially since the authors have not modeled the missing data mechanism.
>
> Imputing representations through posterior distributions is an ideal way of generating missing representations.
> We follow this strategy and formulate the missing data mechanism as a data generation problem through a probabilistic model (i.e., the joint distribution and its corresponding marginals as generators).
> Without loss of generality, we adopt the unconditional parameterization of $\boldsymbol \mu$ and $\boldsymbol \Sigma$ where the gradients are backpropagated during training.
> In this sense, the means and covariance matrices of the GMMs contain not only the label information but also the structural interactions modelled by copulas.
> Hence, the resulting marginal GMMs can generate accurate representations for missing modalities.
>
> >>What does \( q(\theta) \) represent on line 224? Additionally, the connection between \( f_\Theta(x) \) and the copula distribution is not clarified.
> ### Connection between $f_\Theta$ and Copula Distribution.
> The notation $\theta$ denotes the copula parameter, which inherently represents the correlation/interactions between the modalities, while
> $\Theta$ represents the neural network weights.
> To avoid confusion, we have changed $\theta$ to $\alpha$ in the revised manuscript.
> We have also revised the notations in the algorithms for better presentation:  $\hat{y_i}$ should be just obtained from $f_\Theta(\boldsymbol x_1^{(i)}, \ldots, \boldsymbol x_M^{(i)})$ and there should be no posterior distribution for the neural network weights $\Theta$ (i.e., no $q(\Theta)$)
> ### Typos in the manuscripts.
> Thank you for pointing out the typos in the manuscript.
> The $c_M$ you pointed out should be $u_M$ which indicates the quantile of the $M$-th modality.
> We have performed a thorough check and corrected the typos in the revised manuscript.

---

> > ### Comment · Reviewer_Zvwa · 2024-11-24
> > **additional comments.**
> >
> > if the goal is to integrate multiple modalities to improve prediction, accurately modeling the joint distribution of these modalities may not be necessary
> >
> > ****** Your response does not address my point on the importance of modeling the joint distribution of multi-modalities. To me, it is not necessary. Say, Y=f(X, Z), where X is a set of clinical variables and Z is medical imaging. f(., .) is often approximated by using various deep learning methods, such as RNN. I believe that you do not need to model their joint distribution in order to improve the prediction accuracy.
> >
> > It is unclear how a Gaussian mixture distribution for each modality generates accurate representations for missing modalities, especially since the authors have not modeled the missing data mechanism.
> >
> > **********Hence, the resulting marginal GMMs can generate accurate representations for missing modalities.
> >
> > I did not see your point that you can use GMM to generate better missing modalities. You did not model missing mechanisms so that you do not know whether the missing modalities were caused by some unknown machnisms. You overstate what your model can do.

---

> ### Author Response · Authors · 2024-11-25
> **Thank you for your reply.**
>
> ### **[Q1] Joint Distribution**
> > Your response does not address my point on the importance of modeling the joint distribution of multi-modalities. To me, it is not necessary. Say, $Y=f(X, Z)$, where X is a set of clinical variables and Z is medical imaging. f(., .) is often approximated by using various deep learning methods, such as RNN. I believe that you do not need to model their joint distribution in order to improve the prediction accuracy.
>
> Thank you for your reply.
>
> We would like to emphasize that we do not claim it is **necessary** to learn the joint distribution in multimodal learning, while we propose it as a **more promising** approach to handling multimodal learning problems.
> Our focus throughout the paper is to demonstrate the effectiveness of copula models in facilitating multimodal learning.
> Our numerical experiments also demonstrate good performance using such a probabilistic approach with copula.
>
>
> Back to your suggested example:
> > Say, $Y=f(X, Z)$, where X is a set of clinical variables and Z is medical imaging. f(., .) is often approximated by using various deep learning methods, such as RNN.
>
> This is a **general** formulation of multimodal learning, where almost all methods (including ours) follow this standard pipeline.
> Specifically, we adopt **the middle fusion mechanism** in this work --- we first **encode** raw data input $X$ and $Z$ into latent features, say $V_1$ and $V_2$, and then **fuse** them into a joint/fused latent representation $V$.
> Finally, we make a prediction through an output layer (e.g., MLP+softmax) to predict $Y$.
> Under a probabilistic framework, this is equivalent to finding the posterior distribution $Y | X, Z$ or $f_{Y | X, Z}(y|x, z)$ through the use of latent variable $V$ (i.e., we first learn $V | X, Z$ through encoding, and then make predictions through $Y | V$).
>
>
> During the **encoding** phase, we adopt the standard architectures (e.g., LSTM) to encode $X$ and $Z$ into latent embeddings.
> However, when **fusing** the representations $V_1, \ldots, V_M$ into $V$, the notion of **joint distribution** would become prominent.
> As we mentioned, traditional methods use naive concatenations or empirical alignment (e.g., through transformers) to learn the distribution of $V$, which may not approximate the distribution or representation of $V$ well, especially when the data is heavy-tailed (i.e., the tail risk is significant).
> Hence, we proposed that modeling the joint distribution can **largely (if not necessarily) enhance** the predictive performance of $Y$ as the joint distribution accounts not only the intra-modal (i.e., through marginals) but also the inter-modal information.
> As this adopted fundamental pipeline is standard and well-known in multimodal learning, we only mentioned it briefly in the main manuscript and cited the related works to avoid the loss of focus.
>
>
> We agree with you that there could be other approaches to model data of multi-modalities, where using RNN to model $Y$ is certainly viable.
> Our proposal is to take a **distributional approach** to model the joint distribution of **latent representations** to obtain a better distribution of $V$, such that the prediction $Y|V$ based on $V$ can be more accurate.
> Because we model both the marginal and joint distributions, our approach can be viewed as a probabilistic method by aligning different modalities at the distribution level [1].
> This also renders desirable interpretations with marginal effects (i.e., intra-modal effects) and joint effects (i.e., inter-modal effects).
> Copula, although well-established in statistics, is still less developed in machine learning, especially multimodal learning.
> We hope our approach could generate more interest in our research community to explore more potential usage of the copula structures.
>
> > [1] Nedelkoski, Sasho, Mihail Bogojeski, and Odej Kao. "Learning more expressive joint distributions in multimodal variational methods." Machine Learning, Optimization, and Data Science: 6th International Conference, LOD 2020, Siena, Italy, July 19–23, 2020, Revised Selected Papers, Part I 6. Springer International Publishing, 2020.

---

> ### Author Response · Authors · 2024-11-25
> **Thank you for your reply**
>
> ### **[Q2] Missing Mechanism**
> > I did not see your point that you can use GMM to generate better missing modalities. You did not model missing mechanisms so that you do not know whether the missing modalities were caused by some unknown mechanisms. You overstate what your model can do.
>
> Thank you for clarifying the definition of the missing mechanism. We agree that one may leverage the observed missing mechanism in a specific problem domain (e.g., patients of certain traits have missing CXR images) to tackle the missing data problem.
> However, as we target general multimodal learning and have no prior knowledge of the missing mechanism, we **do not** (or cannot) assume a specific pattern/mechanism for the missing modality.
> Hence, we only adopt the most general assumption of missing modality, i.e., as we specified in line 249 of our manuscript,
>
> >      line 249: Without loss of generality, we consider missing modalities with complete labels where only the observations are missing.
>
> Under this assumption, our strategy remains viable as we adopt a **middle fusion strategy** and perform imputation at the **feature level** (not on raw inputs $X$ and $Z$).
> In particular, without assuming the mechanism/pattern of the missing modality, we can still devise an imputation strategy on **feature level** for the missing modality. This strategy is also adopted by other works (e.g., [2-4]).
> From a probabilistic perspective, missing modalities can be treated as unknown variables, which can be naturally imputed using the data generation mechanism estimated by the GMM.
> With the help of copula and the GMM assumption, we can maximize the usage of intra- and inter-modal information to impute the missing observations.
>
>
> In summary, considering that we only impute the latent representations, we believe that we do not overstate the capability of our imputation module, as it optimally utilizes the available information within the general context of multimodal learning.
>
> >[2] Tran, Luan, et al. "Missing modalities imputation via cascaded residual autoencoder." Proceedings of the IEEE conference on computer vision and pattern recognition. 2017.
>
> >[3] Ma M, Ren J, Zhao L, et al. Smil: Multimodal learning with severely missing modality[C]//Proceedings of the AAAI Conference on Artificial Intelligence. 2021, 35(3): 2302-2310.
>
> >[4] Wang, Hu, et al. "Multi-modal learning with missing modality via shared-specific feature modelling." Proceedings of the IEEE/CVF Conference on Computer Vision and Pattern Recognition. 2023.

---

> > ### Comment · Reviewer_Zvwa · 2024-11-26
> > **missing data mechanism**
> >
> > When dealing with missing data, it is necessary to make implicit or explicit assumptions about the mechanisms underlying the missingness. In your case, you essentially assume MCAR or MAR and use the general correlation structure to impute the missing data. However, for prediction, it is evident that imputing missing modalities can introduce biases.

---

> > > ### Author Response · Authors · 2024-11-27
> > > **Further response to missing data mechanism**
> > >
> > > From the Bayesian perspective, missing data are treated equally as unknown parameters, i.e., we can sample the posteriors of missing data based on the full conditional distributions. We assume MAR in our setting, so that the posterior distributions of missing data can be easily obtained using the fitted GMM model. Moreover, the multimodalities are not completely missing, i.e., we are dealing with partially matched modalities. The GMM can be well learned through those matched modalities which in turn can be used to learn the missing modalities. In spirit, this is similar to the Bayesian data augmentation, which is a well-established area in Bayesian missing data problems. Under the MAR assumption, our method would not introduce bias. In addition, similar procedures have also been investigated in-depth by the cited papers in the machine learning literature, where they assume MAR without formal specification. Please see the literature below for similar approaches.
> > >
> > > > [1] Wu, Zhenbang, et al. "Multimodal patient representation learning with missing modalities and labels." The Twelfth International Conference on Learning Representations. 2024.
> > >
> > > > [2] Tran, Luan, et al. "Missing modalities imputation via cascaded residual autoencoder." Proceedings of the IEEE Conference on Computer Vision and Pattern Recognition. 2017.
> > >
> > > > [3] Ma M, Ren J, Zhao L, et al. Smil: Multimodal learning with severely missing modality[C]//Proceedings of the AAAI Conference on Artificial Intelligence. 2021, 35(3): 2302-2310.
> > >
> > > > [4] Wang, Hu, et al. "Multi-modal learning with missing modality via shared-specific feature modelling." Proceedings of the IEEE/CVF Conference on Computer Vision and Pattern Recognition. 2023.
> > >
> > >
> > >
> > > We emphasize that our setting is different from the typical missing data problem in statistics where either the covariates or response are missing. Our missing modality refers to the latent representations, i.e., they are in the latent space and unobserved. We are not imputing any covariates or response which are typically modeled in statistics. The context you are talking about is very different from our setting, and we believe this has caused the confusion.

---

> ### Author Response · Authors · 2024-12-02
>
> Dear reviewer Zvwa,
>
> Thanks for your efforts and thoughtful reviews.
> Since the reviewer-author discussion session will end in a few days, it would be great if you could check our responses to your concerns and let us know if there are any further questions or remaining concerns. We will happily respond if so. Also, we would really appreciate it if you could consider increasing the score if we have made satisfactory responses to your questions.
> Thanks again for your valuable contributions to this process!
>
> Sincerely,
>
> CM$^2$ authors

---

### Official Review · Reviewer_9Psp · 2024-11-04

**Soundness:** 4
**Presentation:** 4
**Contribution:** 3
**Rating:** 8
**Confidence:** 4

**Summary:**

This paper presents a multimodal framework by aligning the modality embedding with copula following a VAE-like structure. Overall I think this paper is a solid paper. The study is comprehensive with detailed equations and experiment setup. The provided code is clean and includes all details.

**Strengths:**

This paper has several strengths:
1: The equations and theorems are clear and easy to follow.
2: The ablation study is extensive.
3: The provided code is clean and wel-organized. Although I haven't run it, it is easy to follow and results seem reproducible.

**Weaknesses:**

I have several questions after reading the experiments:
1: As part of the claimed contribution in this paper is to tackle the missingness, I think a common scenario in healthcare should be discussed - when certain variables or chunks of time series are missing. How can copula deal with the partially observed EHR tabular data, or time series?
2: I think the baselines are not comprehensive. There are many multimodal works in the past year, but there were only 5 baselines and 4 of them were published more than two years ago. In particular, I think the lack of multimodal LLM and/or how copula can be applied in multimodal LLM would be very important.

**Questions:**

The authors did not evaluate the choice of backbone encoders, which I think should be a key lever in the experiment results. Can authors share more insights on the candidate backbone encoders and how much they differed in performance?

---

> ### Author Response · Authors · 2024-11-21
> **We greatly appreciate your positive comments on our manuscript and your insightful summary of the impacts and novelty of our work.**
>
> We greatly appreciate your positive comments on our manuscript and your insightful summary of the impacts and novelty of our work.
> We take great care in responding to several intriguing discussions you raised as follows:
> >>As part of the claimed contribution in this paper is to tackle the missingness, I think a common scenario in healthcare should be discussed - when certain variables or chunks of time series are missing. How can copula deal with the partially observed EHR tabular data, or time series?
>
>
> Thank you for the interesting question. Since we adopt a GMM to model the intra-model distribution, each mixture component can capture the specific characteristic of the data.
> Therefore, even if an intra-modality data point is partially observed (e.g., some chunks are missing), these data can be imputed or recovered by the mixture components capturing the characteristics of those missing partitions.
> This scenarios would be incorporated in future developments of our method.
>
> ### More Competitive Baselines.
> Thank you for your suggestions. We have incorporated a comparison with two recent baselines, specifically LSMT(2023) [1] and Interleaved(2023) [2].
> LSMT is a transformer-based model designed for the multimodal medical context, while Interleaved is a multimodal approach that addresses the irregularity of medical multimodal data and fuses representations from different modalities using cross-modal attention.
> We provide the results of these additional baselines on the MIMIC-IV dataset below. The results indicate that both baselines perform worse than our method on both the fully matched and partially matched datasets. However, the performance gap narrows on the partially matched dataset, suggesting that these baselines are more sensitive to dataset scale, as the partially matched dataset contains a significantly larger number of observations.
>
> | Model   | IHM    |        | READM  |        |
> | ------  | ------ | ------ | ------ | ------ |
> |         | AUROC  | AUPR   | AUROC  | AUPR   |
> | Totally Matched |
> | LSMT    | $0.803_{(0.769, 0.837)}$    | $0.444_{(0.370, 0.519)}$   | $0.701_{(0.662, 0.737)}$  | $0.421_{(0.356, 0.490)}$  |
> | Interleaved| $0.800_{(0.764, 0.834)}$ | $0.440_{(0.374, 0.523)}$   | $0.702_{(0.664, 0.741)}$  | $0.421_{(0.360, 0.487)}$  |
> | $CM^2$  | $0.827_{(0.790, 0.859)}$    | $0.492_{(0.423, 0.566)}$  | $0.737_{(0.704, 0.773)}$  | $0.466_{(0.404, 0.529)}$  |
> |Parially Matched
> | LSMT    | $0.854_{(0.838, 0.870)}$    | $0.508_{(0.466, 0.551)}$  | $0.764_{(0.746, 0.781)}$  | $0.473_{(0.436, 0.509)}$  |
> | Interleaved| $0.856_{(0.840, 0.871)}$ | $0.508_{(0.466, 0.550)}$   | $0.758_{(0.740, 0.775)}$  | $0.473_{(0.441, 0.506)}$  |
> | $CM^2$  | $0.858_{(0.844, 0.872)}$    | $0.527_{(0.490, 0.568)}$   | $0.771_{(0.752, 0.788)}$  | $0.486_{(0.452, 0.518)}$  |
>
>
> >[1] Firas Khader, Gustav Müller-Franzes, Tianci Wang, Tianyu Han, Soroosh Tayebi Arasteh, Christoph Haarburger, Johannes Stegmaier, Keno Bressem, Christiane Kuhl, Sven Nebelung, et al. Multi-modal deep learning for integrating chest radiographs and clinical parameters: a case for transformers. Radiology, 309(1):e230806, 2023
>
> >[2] Xinlu Zhang, Shiyang Li, Zhiyu Chen, Xifeng Yan, and Linda Ruth Petzold. Improving medical predictions by irregular multimodal electronic health records modeling. In International Conference on Machine Learning, pp. 41300–41313. PMLR, 2023.13

---

> > ### Author Response · Authors · 2024-11-21
> >
> > ### Effects with Different Backbones.
> > We are aware that the choice of backbone encoders may influence the performance of all methods. To address this, we conducted additional experiments to evaluate the impact of different encoder architectures for each modality. Specifically, we used the Transformer and ViT as alternative backbone encoders for the time-series and CXR image data, respectively.
> > The results of these experiments, conducted on the MIMIC-IV fully matched dataset, are presented below. Additionally, we provide results with 95\% confidence intervals in Table 9 in the Appendix of the revised manuscript.
> > We observe that our method consistently outperforms competitive baselines across various backbone encoders, highlighting its robustness and effectiveness. Furthermore, our method demonstrates greater stability across different backbones, suggesting it is less sensitive to their selection.
> > Besides, the Transformer backbone generally outperforms the LSTM backbone, particularly for  MMTM, LSMT, and Interleaved.
> > While the ResNet backbone slightly outperforms the ViT backbone, the performance difference is not substantial, suggesting time-series data's greater impact on backbone encoder choice.
> >
> > | Model   | Encoder\_TS | Encoder\_I | IHM    |        | READM  |        |
> > | ------  | ----       | ------    | ------ | ------ | ------ | ------ |
> > |         |            |           | AUROC  | AUPR   | AUROC  | AUPR   |
> > |         | LSTM       |  ResNet   |        |        |        |        |
> > | MMTM    |            |           | 0.802  | 0.429  | 0.713  | 0.420  |
> > | DAFT    |            |           | 0.815  | 0.454  | 0.729  | 0.433  |
> > | Unified |            |           | 0.808  | 0.429  | 0.719  | 0.450  |
> > | MedFuse |            |           | 0.813  | 0.448  | 0.725  | 0.438  |
> > | DrFuse  |            |           | 0.814  | 0.450  | 0.723  | 0.422  |
> > | LSMT    |            |           | 0.803  | 0.444  | 0.701  | 0.421  |
> > | Interleaved|         |           | 0.800  | 0.440  | 0.702  | 0.421  |
> > | $CM^2$(Ours) |       |           | 0.827  | 0.492  | 0.737  | 0.466  |
> > |         | LSTM       |  ViT      |        |        |        |        |
> > | MMTM    |            |           | 0.805  | 0.446  | 0.712  | 0.422  |
> > | DAFT    |            |           | 0.808  | 0.438  | 0.714  | 0.423  |
> > | Unified |            |           | 0.803  | 0.431  | 0.707  | 0.416  |
> > | MedFuse |            |           | 0.805  | 0.439  | 0.715  | 0.424  |
> > | DrFuse  |            |           | 0.806  | 0.446  | 0.716  | 0.421  |
> > | LSMT    |            |           | 0.801  | 0.441  | 0.703  | 0.410  |
> > | Interleaved|         |           | 0.802  | 0.434  | 0.710  | 0.435  |
> > | $CM^2$(Ours) |       |           | 0.826  | 0.490  | 0.736  | 0.452  |
> > |         | TF         |  ResNet   |        |        |        |        |
> > | MMTM    |            |           | 0.813  | 0.452  | 0.735  | 0.448  |
> > | DAFT    |            |           | 0.814  | 0.437  | 0.730  | 0.430  |
> > | Unified |            |           | 0.812  | 0.453  | 0.719  | 0.426  |
> > | MedFuse |            |           | 0.815  | 0.441  | 0.728  | 0.442  |
> > | DrFuse  |            |           | 0.818  | 0.460  | 0.726  | 0.430  |
> > | LSMT    |            |           | 0.817  | 0.452  | 0.722  | 0.431  |
> > | Interleaved|         |           | 0.821  | 0.459  | 0.721  | 0.429  |
> > | $CM^2$(Ours) |       |           | 0.823  | 0.488  | 0.740  | 0.470  |
> > |         | TF         |  ViT      |        |        |        |        |
> > | MMTM    |            |           | 0.813  | 0.462  | 0.723  | 0.435  |
> > | DAFT    |            |           | 0.803  | 0.432  | 0.719  | 0.421  |
> > | Unified |            |           | 0.812  | 0.463  | 0.719  | 0.412  |
> > | MedFuse |            |           | 0.818  | 0.461  | 0.721  | 0.431  |
> > | DrFuse  |            |           | 0.814  | 0.436  | 0.717  | 0.416  |
> > | LSMT    |            |           | 0.815  | 0.453  | 0.714  | 0.424  |
> > | Interleaved|         |           | 0.818  | 0.453  | 0.717  | 0.433  |
> > | $CM^2$(Ours) |       |           | 0.826  | 0.489  | 0.737  | 0.465  |
> >
> >
> > >>In particular, I think the lack of multimodal LLM and/or how copula can be applied in multimodal LLM would be very important.
> >
> > Thank you for your interesting suggestions.
> > We agree that application to large multimodal models is a promising field.
> > The copula alignment can be adopted similarly when training the multimodal LLM, where learning the joint distributions of the image and text would be incorporated in contrastive pre-training (e.g., [3], which introduces a probabilistic model (Gaussian) for VLM training).
> > We are actively exploring the application of copula to multimodal language models, especially vision language models in our ongoing works.
> > >[3] Upadhyay, Uddeshya, et al. "Probvlm: Probabilistic adapter for frozen vison-language models." Proceedings of the IEEE/CVF International Conference on Computer Vision. 2023.

---

### Official Review · Reviewer_F7PH · 2024-11-04

**Soundness:** 3
**Presentation:** 3
**Contribution:** 2
**Rating:** 6
**Confidence:** 3

**Summary:**

This paper proposed a copula-based model to model interactions among modalities. Gaussian mixture model was firstly employed to learn the marginal distribution of each component. After that, the parameters of the copula-based model, and of the classifier are jointly optimized via variation inference. Experiments are conducted on MIMIC dataset.

**Strengths:**

This paper is well organized and written. The idea of applying Copula model to model interactions is interesting.

**Weaknesses:**

My main concerns are about the Copula model. First, according to theorem 1, once the marginal distribution of each component F_i(z) is provided (we first assume we can learn it well), there exits a copula C to recover the joint distribution with C(F1(z1),F2(z2),...,Fm(zm)). That means, we should parameterize q(z1,...,zm) as C(F1(z1),...,Fm(zm)) or C(z1,...,zm)\pi_{m=1}^M Fm(zm). But in your variational family, the form of q(z) seems inconsistent with the above forms. Second, why do we assume GMM to fit the marginal distribution? Have you performed goodness of test to show that latent z are mixture of Gaussian? Empirically, how do we choose K in the GMM?

The experimental result is not convincing enough, since the proposed method does not significantly outperform others. Specifically, the CI of these methods are very overlapped, making it hard to demonstrate the improvement.

Moreover, although this idea is interesting, this paper seems a simple application of the copula model. From my perspective, the novelty does not reach the acceptance bar of ICLR.

**Questions:**

Where is the definition of \theta? From my current understanding, it refers to the parameter of classifiers. But according to line 249, it also includes parameters in the embedding (encoder). If that is the case, then the encoded latent variables will change after each iteration, then is that mean we should apply GMM to learn \mu and \Sigma in each iteration?

**Details Of Ethics Concerns:**

Not applicable.

---

> ### Author Response · Authors · 2024-11-21
> **We thank the reviewer for your valuable comments on our manuscript.**
>
> We thank the reviewer for your valuable comments on our manuscript.
> We will address your comments on the significant experiments, the GMM assumption, and the novelty of our work as follows.
> >>My main concerns are about the Copula model. First, according to theorem 1, once the marginal distribution of each component $F_i(z)$ is provided (we first assume we can learn it well), there exits a copula C to recover the joint distribution with $C(F_1(z_1),F_2(z_2),...,F_M(z_M))$. That means, we should parameterize $q(z_1,...,z_M)$ as $C(F_1(z_1),F_2(z_2),...,F_M(z_M))$ or $C(z_1,...,z_M)\prod_{m=1}^M F_m(z_m)$. But in your variational family, the form of $q(\boldsymbol z)$ seems inconsistent with the above forms.
>
> We appreciate your in-depth observations and we would like to further clarify to avoid potential misunderstanding.
> For the variational family, we use the joint **density** distribution instead of the **cumulative** distribution to obtain the log-likelihood.
> In fact, if one takes the partial derivatives of Equation (2) with respect to $x_1, \ldots, x_M$ (or $\boldsymbol z_1, \ldots, \boldsymbol z_M$ in our notation), they can obtain the joint density introduced by $q(\boldsymbol z)$ in the variational family.
> The variational objective is seemingly defined in [1], where its correctness can be verified.
> >[1] Tran, Dustin, David Blei, and Edo M. Airoldi. "Copula variational inference." Advances in neural information processing systems 28 (2015).
>
> >>Second, why do we assume GMM to fit the marginal distribution? Have you performed goodness of test to show that latent z are mixture of Gaussian?
>
> We discussed the reasons of assuming a Gaussian mixture model in the general response.
> The goodness-of-fit test in high dimensions is still a developing field in statistics and there is little research in this direction due to its formidable challenges.
> Even for VAE, the validity of assuming a Gaussian prior is still under intense discussion.
> We adopt a Gaussian mixture model in light of the many empirical successes of assuming a GMM on the latent feature distributions.
> With the future advances in statistical theory in high-dimensional goodness-of-fit tests, we would attempt to validate the theoretical soundness of the GMM assumptions.
>
> >>Empirically, how do we choose K in the GMM?
>
> As a convention in statistical modelling, $K$ is set to be small to avoid over-specification.
> The popular choice of $K$ is 2 to 3 such that the learned mixture distribution will achieve an optimal degree of flexibility while preventing over-specification.
> In the additional experiments (presented in Section F of the Appendix), we evaluate how the performance changes with the changes of $K$.
> We observe that the performance is quite robust.

---

> > ### Author Response · Authors · 2024-11-21
> >
> > >>The experimental result is not convincing enough, since the proposed method does not significantly outperform others. Specifically, the CI of these methods are very overlapped, making it hard to demonstrate the improvement.
> >
> > We are aware that there are some overlaps in the confidence intervals, which due to a high confidence level (i.e., 95\%) which specifies a wide range of possible performance.
> > However, the upper and lower quantiles of the CI are generally outperforming the baselines.
> > To demonstrate that our results are still significant, we perform two-sample bootstraping $t$-test to justify.
> > Table 10 in the Appendix of the revised manuscript demonstrates the results of the student's $t$-test on the MIMIC-IV dataset.
> > We observe that the improvements over the competitive baselines are overall statistically significant under the 0.05 significance level, validating the effectiveness of our method.
> >
> > | Model   | IHM    |        | READM  |        |
> > | ------  | ------ | ------ | ------ | ------ |
> > |         | AUROC  | AUPR   | AUROC  | AUPR   |
> > | $CM^2$ vs. MMTM | $2.02 \times 10^{-6}$ | $3.55 \times 10^{-180}$ | $4.40\times 10^{-100}$ | $5.36\times 10^{-291}$ |
> > | $CM^2$ vs. DAFT | $0.1122$ | $1.53 \times 10^{-132}$ | $9.37\times 10^{-78}$ | $2.95\times 10^{-240}$ |
> > | $CM^2$ vs. Unified | $4.55\times 10^{-08}$ | $5.71\times 10^{-240}$ | $4.80\times 10^{-73}$ | $2.81\times 10^{-139}$ |
> > | $CM^2$ vs. MedFuse | $7.73\times 10^{-07}$ | $5.66\times 10^{-129}$ | $1.11\times 10^{-92}$ | $3.69\times 10^{-173}$ |
> > | $CM^2$ vs. DrFuse | $0.1447$ | $4.28\times 10^{-99}$ | $6.05\times 10^{-67}$ | $6.25\times 10^{-250}$ |
> >
> >
> > >>Moreover, although this idea is interesting, this paper seems a simple application of the copula model. From my perspective, the novelty does not reach the acceptance bar of ICLR.
> >
> > We observe that modelling the dependence or interaction structure is important in multimodal learning
> > The major motivation of our work is to borrow the capability of copula to model complex interactions between the variables, which suits the key issues in multimodal learning in aligning the distributions of modalities.
> > Our work is a pioneer work bridging the extensive research in copula modelling and multimodal learning, and facilitating the future development of more powerful multimodal learning algorithms.
> > This contribution is also acknowledged by reviewer 9Psp, as copula has been under-explored in machine learning.
> > With this pioneer work, more advanced copula, such as vine copula can be introduced in future works to customize the dependence/interaction structure according to the domain knowledge.
> > Additionally, our model can better address missing modality imputation with the proposed probablistic generator, since the imputed features contain
> > the inter-modal information encoded by copula parameter, and the intra-model information learned by GMM marginals.
> >
> > On the other hand, the **tail dependence** is typically not evident in most of the multimodal data, leading to difficulty in estimating the tail dependence empirically (since there is little observations with extreme values).
> > Hence introducing copula models (e.g.,  the Gumbel, which is formulated based on the extreme values of the assumed distributions) can effectively capture the tail dependence, which cannot be estimated from most of the previous methods.
> >
> > >>Where is the definition of $\theta$? From my current understanding, it refers to the parameter of classifiers. But according to line 249, it also includes parameters in the embedding (encoder). If that is the case, then the encoded latent variables will change after each iteration, then is that mean we should apply GMM to learn $\mu$ and $\Sigma$ in each iteration?
> >
> > We apologize for the potential confusion.
> > We define $\theta$ as the copula modality which specifies the degree of intercorrelations between the modalities, while the bigger $\Theta$ represents the parameters of the encoders together with the classifiers.
> > We agree that the notations may be confusing.
> > Therefore we have changed the copula parameter $\theta$ as $\alpha$ in the revised manuscript.
> > Furthermore, we backpropagate the label information together with the likelihood to $\boldsymbol \mu$ and $\boldsymbol \Sigma$, together to the encoder weights, after each iteration.

---

> > > ### Comment · Reviewer_F7PH · 2024-11-22
> > > **Update**
> > >
> > > Thank you for your response. Most of my concerns are addressed. I have increased the score from 3 to 6.

---

> > > > ### Author Response · Authors · 2024-11-27
> > > >
> > > > Thank you for your appreciation of our response!

---

### Official Review · Reviewer_haPi · 2024-11-06

**Soundness:** 3
**Presentation:** 4
**Contribution:** 3
**Rating:** 6
**Confidence:** 4

**Summary:**

This paper addresses the missing modality problem in multimodal machine learning for healthcare using the statistical copula model. The copula model is powerful in modeling the interactions between random variables by directly describing the dependency structures between them. This paper assumes a Gaussian mixture distribution for the marginals (each modality) and uses the copula for the joint distribution. Given the probabilistic nature of the model, when a modality is missing, sampling can be performed. Therefore, the problem of missing modality and modeling of cross-modal interaction are jointly tackled. Empirical evaluations show promising performance in comparison with existing approaches.

**Strengths:**

- The paper is well-written and easy to follow. I find it quite pleasant to read the paper.
- The proposed method is technically sound. Using a copula to capture the joint distribution is a simple yet effective way of modeling the cross-modal interactions.
- Empirical evaluation shows good performance.

**Weaknesses:**

- The theoretical analysis is not in-depth. Section 4.5 presents an existing theorem to justify the uniqueness of the joint distribution, but its implication to the multimodal learning problem is not analyzed clearly and in-depth.
- The marginals are assumed to be mixtures of Gaussian. This assumption seems to lack justification and no alternatives to GMM is discussed.

**Questions:**

- How to interpret Fig. 2? Why is the Gumbel copula more focused on the positive dependence between the modalities while the Gaussian copula has less weight on modeling tail dependences?
- In Table 5, the effect of different copula families does not affect the performance for the partially matched dataset, what are the reasons behind?
- In Table 5, for a matched subset, copula families make quite a difference in the performance. How should the users choose the correct copula family to use for their own dataset?

---

> ### Author Response · Authors · 2024-11-21
> **We greatly appreciate your positive comments on the soundness and effectiveness of our method.**
>
> We greatly appreciate your positive comments on the soundness and effectiveness of our method.
> We take great care in responding to several intriguing discussions raised by you as follows:
> >>The theoretical analysis is not in-depth. Section 4.5 presents an existing theorem to justify the uniqueness of the joint distribution, but its implication to the multimodal learning problem is not analyzed clearly and in-depth.
>
> We agree that theoretical insights or properties would be beneficial in understanding the soundness of applying copula to multimodal learning.
> The key focus of our work is to methodologically formulate the copula multimodal learning, with most of the discussion on how to incorporate copula to large-scale multimodal learning problems (with variational inference).
> Based on Sklar's theorem, the copula is a well-established method in statistics to model a joint distribution using marginal distributions (which can be viewed as multiple modalities). However, the copula framework is under-explored in machine learning, although it naturally fits to the paradigm of multimodal learning.
> The main **implication of Sklar's theorem** is that it allows us to update the copula parameter and the marginal distributions separately.
> Under the **multimodal learning context**, such properties allow us to learn the intra-modal information (through learning the marginals) and inter-modal information (through learning the copula parameter) in parallel, which avoids the collision in information and optimizes the learning outcome.
> The development of theoretical properties of copula multimodal deep learning requires a significant amount of work which warrants further comprehensive investigation
> in future works.
> >>The marginals are assumed to be mixtures of Gaussian. This assumption seems to lack justification and no alternatives to GMM is discussed.
>
> We discussed the reasonability of choosing the Gaussian mixture model (GMM) as the marginal distribution in the general response.
> Since the development of a high dimensional distribution is rare, we select the most common assumption (i.e., GMM) that presents the greatest degree of flexibility in modeling high dimensional data.
> Assuming more complex distributions other than multivariate Gaussian or GMM poses the risk of mis-specification, as these distributions mostly impose additional assumptions on the feature space.
>
> We present a potential alternative by assuming only a multivariate Gaussian distribution in the revised manuscript (commonly assumed by VAE works).
> We observe that the performance in general worsens as the assumption is too strong and lacks flexibility compared to GMM.
> >>How to interpret Fig. 2? Why is the Gumbel copula more focused on the positive dependence between the modalities while the Gaussian copula has less weight on modeling tail dependences?
>
>
> Figure 2 presents the learned joint distribution under different assumptions of the copula family.
> Overall, we observe that the learned interactions are all positively correlated (i.e., slope > 0), indicating that all families capture a similar pattern of dependence.
> However, the pattern at tails would be slightly different due to the unique definitions of families.
> We present more discussions in the Appendix on how the copula distribution changes as the training epoch increases.

---

> > ### Author Response · Authors · 2024-11-21
> >
> > ### Difference Between Gaussian and Gumbel Copulas.
> > Gaussian copula is that it does not have tail dependence, and hence the Gaussian copula will underestimate the probability of joint extreme events [1].
> > On the other hand, the Gumbel copula is formulated based on the extreme values of the assumed distributions, and hence it can better capture the tail dependence.
> > If the tail dependence is evident at the feature level, then we fit copula models (e.g., Gumbel) that are believed to possess matching tail dependence properties.
> > >[1] Joe, Harry. Dependence modeling with copulas. CRC press, 2014.
> >
> > >>In Table 5, the effect of different copula families does not affect the performance for the partially matched dataset, what are the reasons behind?
> > >>In Table 5, for a matched subset, copula families make quite a difference in the performance. How should the users choose the correct copula family to use for their own dataset?
> > ### Difference in Table 5 Performances.
> > Due to the limited number of observations, the performance across copula families tends to be more variable in the matched subset .
> > On the other hand, the observations are more sufficient in the partially matched datasets, leading to relatively stable performance across families.
> > This demonstrates the importance of choosing a correct copula family since the tail risks is more evident as the number of observations decreases.
> > We will further discuss the choice of copula family in the next section.
> > ### Choice of the Copula Family.
> > A straightforward way is that we can compute the likelihood during the validation process and select the copula family with the highest likelihood.
> > Additionally, we can estimate the upper and lower tail dependence empirically to choose the best-matching copula family;
> > for instance, using the empirical estimator based on multivariate extreme-value theory developed by Huang [2].
> > By matching the empirical tail dependence to the tail dependence obtained by the respective copula families, we can select the optimal copula family matching the tail dependence.
> >
> > Furthermore, we focused on some classical symmetric copula families in this work.
> > Since copula is a large research field in statistics, one may customise their own assumption of the interactions between the modalities using a more advanced copula structure (e.g., vine copula [3], which constructs a $d$-dimensional graphical model to model the interactions) to tackle more complex multimodal learning problems.
> > We will introduce more advanced copula families in future works with more focus on modelling the tail dependence/tail risks of the data.
> > >[2] Huang, Xin. Statistics of bivariate extreme values. Amsterdam: Thesis Publishers, 1992.
> >
> > >[3] Nagler, Thomas, Daniel Krüger, and Aleksey Min. "Stationary vine copula models for multivariate time series." Journal of Econometrics 227.2 (2022): 305-324.

---

> ### Comment · Reviewer_haPi · 2024-11-29
> **Thank you for the response**
>
> I sincerely appreciate the responses from the authors. My questions are mostly clarified and I will keep my rating unchanged.

---

### Author Response · Authors · 2024-11-21
**General Comments to All Reviewers**

We thank all the reviewers for your time and efforts on our manuscript.
According to the reviewers' comments, we conduct additional experiments to extensively evaluate our method, including adding more baselines, evaluations with more backbones, and ablation studies on the hyperparameters.
Newly added experimental results can be found in the revised manuscript.
To avoid confusion, we have revised the copula parameter from $\theta$ to $\alpha$ to separate its definition from encoder weights $\Theta$.
### The Reasonability of Gaussian Mixture Model Assumptions:
Our model is inherently similar to a VAE framework (also mentioned by reviewer 9Psp), where the original VAE assumes only the multivariate Gaussian distribution [1].
We follow this strategy and assume the features follow a more flexible distribution --- the Gaussian mixture model (GMM).
Moreover, the use of the GMM is a common technique in machine learning to model the behavior of distributions in high dimensions (See [2-5] for non-exclusive examples).
On one hand, this technique provides a great degree of flexibility in modeling the distribution of latent features. On the other hand, the assumption of GMM demonstrates promising empirical performance in many problem domains.
Therefore, we believe it is a reasonable assumption on the marginal distribution of the modalities.
>[1] Kingma, Diederik P., and Max Welling. "Auto-Encoding Variational Bayes." stat 1050 (2014): 1.

>[2] Song, Andrew H., et al. "Morphological prototyping for unsupervised slide representation learning in computational pathology." Proceedings of the IEEE/CVF Conference on Computer Vision and Pattern Recognition. 2024.

>[3] Bai, Junwen, Shufeng Kong, and Carla P. Gomes. "Gaussian mixture variational autoencoder with contrastive learning for multi-label classification." International conference on machine learning. PMLR, 2022.

>[4] Ni, Jingchao, et al. "Superclass-conditional gaussian mixture model for learning fine-grained embeddings." International Conference on Learning Representations. 2021.

>[5] Zhao, Bingchen, Xin Wen, and Kai Han. "Learning semi-supervised Gaussian mixture models for generalized category discovery." Proceedings of the IEEE/CVF International Conference on Computer Vision. 2023.

---

### Note · Authors · 2025-01-24

I have read and agree with the venue's withdrawal policy on behalf of myself and my co-authors.